# MicroCT-based phenomics in the zebrafish skeleton reveals virtues of deep phenotyping in a distributed organ system

Matthew Hur[1], Charlotte A Gistelinck[2], Philippe Huber[1], Jane Lee[1], Marjorie H Thompson[1], Adrian T Monstad-Rios[1], Claire J Watson[1], Sarah K McMenamin[3], Andy Willaert[2], David M Parichy[4], Paul Coucke[2], Ronald Y Kwon[1]*

[1]Department of Orthopaedics and Sports Medicine, University of Washington, Seattle, United States; [2]Center for Medical Genetics, Ghent University, Ghent, Belgium; [3]Biology Department, Boston College, Massachusetts, United States; [4]Department of Biology, University of Virginia, Charlottesville, United States

*For correspondence:
ronkwon@uw.edu

Competing interests: The authors declare that no competing interests exist.

**Abstract** Phenomics, which ideally involves in-depth phenotyping at the whole-organism scale, may enhance our functional understanding of genetic variation. Here, we demonstrate methods to profile hundreds of phenotypic measures comprised of morphological and densitometric traits at a large number of sites within the axial skeleton of adult zebrafish. We show the potential for vertebral patterns to confer heightened sensitivity, with similar specificity, in discriminating mutant populations compared to analyzing individual vertebrae in isolation. We identify phenotypes associated with human brittle bone disease and thyroid stimulating hormone receptor hyperactivity. Finally, we develop allometric models and show their potential to aid in the discrimination of mutant phenotypes masked by alterations in growth. Our studies demonstrate virtues of deep phenotyping in a spatially distributed organ system. Analyzing phenotypic patterns may increase productivity in genetic screens, and facilitate the study of genetic variants associated with smaller effect sizes, such as those that underlie complex diseases.
DOI: https://doi.org/10.7554/eLife.26014.001

## Introduction

Advances in genomic sequencing have revolutionized our ability to identify gene variants that can impact human health, yet our ability to characterize vertebrate phenomes – i.e., to acquire in-depth phenotypic profiles at the scale of the whole organism (*Houle et al., 2010*) – remains limited. The development of vertebrate phenotypic assays that approach the scale and efficiency of genomic technologies hold promise to fundamentally enhance our functional understanding of genes and genomic variation. For instance, they could rapidly accelerate genetic and drug discovery by enabling organism-wide, unbiased analysis of large numbers of phenotypic features. In addition, they could expand our understanding of functional relationships between genes by helping to cluster mutations into common pathways based on similarities in phenotypic signatures. Finally, since our functional understanding of genes and genomic variation is directly coupled to the depth with which we are able to characterize their effects on phenotype, our understanding of gene function is fundamentally limited by the tendency for phenotypic assays to be restricted to a few readouts (*Houle et al., 2010*; *Schork, 1997*; *Bilder et al., 2009*). A better understanding of the biological insights that may be attained by profiling changes in patterns in a large number of phenotypic

features is essential to both guide and drive the development of next-generation phenotyping technologies.

The skeleton is an organ consisting of a large number of tissues distributed throughout the body; thus, it is a prime example of a spatially distributed organ system that may benefit from phenomic approaches. The skeleton comprises bones of different developmental origins (e.g., neural crest vs. mesoderm derived) and modes of ossification (intramembranous vs. endochondral), cellular compositions, and gene expression patterns. Different compartments can be differentially regulated through differences in local mechanical environment, proximity to tissues and organs that exert paracrine control (e.g., at the muscle/bone interface), and variations in vascularization and innervation. Since different mutations often affect different skeletal compartments, it is common practice to perform skeletal phenotyping at multiple skeletal sites. In this context, most efforts to increase the phenotypic content of skeletal assays have focused on increasing the depth of description at a given anatomical location rather than increasing the number of bones/compartments analyzed. In mice, Adams *et al.* (*Adams, 2015*) developed a semi-automated workflow integrating microCT and multi-image cryohistology (*Dyment et al., 2016*; *Hong et al., 2012*) to quantify 15 phenotypic measures in the femur and lumbar spine. Using a high-throughput automated synchrotron-based tomographic microscopy system, Mader *et al.* (*Mader et al., 2015*) quantified 22 different measurements in the mouse femur. In zebrafish, Pardo-Martin *et al.* (*Pardo-Martin et al., 2013*) used automated sample handling and optical projection tomography to acquire high-dimensional phenotypic profiles (~200 measurements) in the craniofacial cartilage of early larvae, representing one of the most ambitious approaches to perform large-scale phenotyping in the skeleton to date. Yet, even in this analysis, traits were derived from only 9 skeletal elements. Further, this method is not readily extendable to bones outside of the craniofacial skeleton, or to adults. Finally, while both the mouse and zebrafish spine are amenable to whole-body microCT imaging (*Buchan et al., 2014*; *Gray et al., 2014*), in-depth phenotyping is usually limited to a few vertebral bodies (*Bouxsein et al., 2010*). In this context, methods to perform in-depth phenotyping in a large number of bones represents a unique class of problems that has not been adequately addressed.

The objective of this study was twofold: (1) to develop microCT-based methods for profiling morphological and densitometric traits at a large number of anatomical sites in the axial skeleton of adult zebrafish, and (2) to assess the benefits of analyzing phenotypic patterns in discriminating mutant populations. Here, we present a supervised segmentation algorithm, FishCuT, and a statistical workflow to test for differences in phenomic patterns in mutant populations. We demonstrate the potential for vertebral patterns to confer heightened sensitivity, with similar specificity, in discriminating mutant populations compared to analyzing individual vertebrae in isolation. To identify phenomic signatures associated with human brittle bone diseases, we perform in-depth phenotyping in two zebrafish mutants, *plod2*[-/-] and *bmp1a*[-/-]. Further, to demonstrate the potential to identify novel axial skeletal mutants associated with human genetic disorders, we phenotype *opallus*, which harbors a mutation in thyroid stimulating hormone receptor (*tshr*) identical to a mutation causing hyperthyroidism in humans. Finally, we develop phenome-based allometric models and show that they are able to discriminate mutant phenotypes otherwise masked by alterations in growth. We have integrated our methods into a software package, FishCuT, whose source code has been deposited on GitHub as a beta release (*Kwon, 2017*); a copy is archived at https://github.com/elifescien-ces-publications/FishCuT.

## Results

### A microCT-based workflow for phenomic profiling of the axial skeleton in adult zebrafish

Due to their small size, zebrafish are conducive to whole-body microCT imaging at high resolution (*Cheng et al., 2011*). Previous studies have demonstrated radiopacity in adult zebrafish vertebrae (*Fisher et al., 2003*) and the feasibility of imaging vertebral morphology in adult zebrafish via microCT (*Buchan et al., 2014*; *Gray et al., 2014*; *Gistelinck et al., 2016a*; *Asharani et al., 2012*; *Spoorendonk et al., 2008*). Moreover, the small size of zebrafish can be exploited by imaging multiple fish simultaneously. For instance, in most cases we imaged two fish at a time, enabling whole spines to be acquired at 21 μm resolution in as little as ~20 min/fish for fish that were ~20 mm in

standard length. Further, in a single proof-of-concept study, we found that up to eight animals could be scanned simultaneously at 21 µm resolution, enabling whole spines to be imaged in an effective scan time of ~5 min/fish. While several practical issues will need to be resolved prior to broad application of such high-density scanning strategies (discussed below), our studies suggest that the primary bottleneck to phenomic analysis in a large number of bones is not the time required for microCT scan acquisition, but the time required for segmentation. Indeed, we found that ~60 min was required to manually segment a single vertebra of an adult fish scanned at 21 µm resolution (~60 image slices) using a user-assisted segmentation tool, a rate that would require >24 hr to segment the ~27 precaudal and caudal vertebrae that comprise the zebrafish vertebral column.

To overcome this barrier, we developed FishCuT, a microCT analysis toolkit that couples supervised segmentation with connectivity analysis to enable computation of descriptors of bone morphology, mass, and mineralization at a large number of anatomical sites in the vertebral column of adult zebrafish (*Figure 1*). To segment vertebral bodies, FishCuT employs a region-growing algorithm which takes user-specified line regions of interest (ROIs) specifying 'seed' locations of vertebral boundaries, and uses them to isolate individual vertebrae. This segmentation is achieved by iteratively growing a separation boundary such that each vertebra is composed of connected components that do not contain voxels from different vertebrae. Once each vertebral body is segmented (*Figure 1B*), a supervised algorithm is used to segment each vertebral body into three skeletal elements: the neural arch (Neur), centrum (Cent), and haemal arch/ribs (Haem) (*Figure 1C and D*). For each skeletal element, four primary measurements are computed: Tissue Mineral Density (TMD, mgHA/cm$^3$), Volume (Vol, µm$^3$), Thickness (Th, µm), and Surface Area (SA, µm$^2$) (*Figure 1E*). In addition to the above measurements, FishCuT computes centrum length (Cent.Le, µm), as well as intra-specimen variation in TMD and thickness (i.e. TMD.sd and Th.sd, respectively [*Bouxsein et al., 2010*; *Burghardt et al., 2008*]) for each skeletal element. For each measure, the 'total' value (e.g., Tot.TMD) within a single vertebra (i.e., across the centrum, haemal arch/ribs, and neural arch) is also computed. For analysis, each combination of element/outcome (e.g., Cent.TMD) is computed as a function of vertebra number. The global test (*Goeman et al., 2006*; *Goeman et al., 2004*), a regression-based statistical test designed for data sets in which many covariates (or features) have been measured for the same subjects, is used to assess whether the pattern across vertebrae is significantly different between groups. This output is provided graphically via a custom R script that computes plots of phenotypic trends (*Figure 1F*), and color codes them to highlight the trends that are statistically significant in the global test. Finally, for each combination of outcome/element, a standard score is computed as the difference between its value in each vertebral body and its mean value across all vertebrae in the control population, divided by the standard deviation across all vertebrae in the control population. These data are arranged into matrix constructs that we have termed 'skeletal barcodes', which can be input into graphing software to generate heat maps to facilitate visualization of phenotypic trends both within individuals and across groups (*Figure 1G*). Currently, 25 different quantities are computed for each vertebra (Cent.TMD, Cent.Th, Cent.Vol, Cent.Le, Cent.SA, Cent.TMD.sd, Cent.Th.sd, Neur.TMD, Neur.Th, Neur.Vol, Neur.SA, Neur.TMD.sd, Neur.Th.sd, Haem.TMD, Haem.Th, Haem.Vol, Haem.SA, Haem.TMD.sd, Haem.Th.sd, Tot.TMD, Tot.Th, Tot.Vol, Tot.SA, Tot.TMD.sd, Tot.Th.sd). The total number of measures that are computed is dependent on the number of vertebrae which are analyzed by the user (i.e., by seeding separation boundaries). Analyzing 24 vertebrae results in 24 × 25 = 600 phenotypic measures, and usually takes <5 min/fish (~100 x faster than manual segmentation). In our studies we analyzed between 22–28 vertebrae, with an average of 24.1 ± 1.4 vertebrae per fish (mean ± SD, n = 34 fish).

To assess segmentation quality, we analyzed a single wildtype zebrafish (strain, AB; standard length, SL: 24.8 mm) using two different approaches – FishCuT and manual segmentation – and used the Dice similarity coefficient (DSC) (*Zou et al., 2004*) to evaluate spatial overlap in segmentations produced by the two approaches. We computed a DSC of 0.932 ± 0.001 (mean ± SD, n = 2 vertebrae) when we excluded pixels with intensities less than the threshold used in the FishCuT analysis, exceeding the value of 0.7 suggested to indicate excellent agreement between manual and automated segmentation approaches (*Zijdenbos et al., 1994*). We computed a mean difference of 3.4 ± 1.2% (mean ± SD, n = 3 fish) in centrum length, a measure that is calculated directly from user-specified 'seed' locations of the vertebral boundaries, when we serially performed primary and secondary scans of the same animal. This suggests that intra-operator reproducibility during this step was high. Finally, to assess independence between phenotypic measures, we computed Pearson

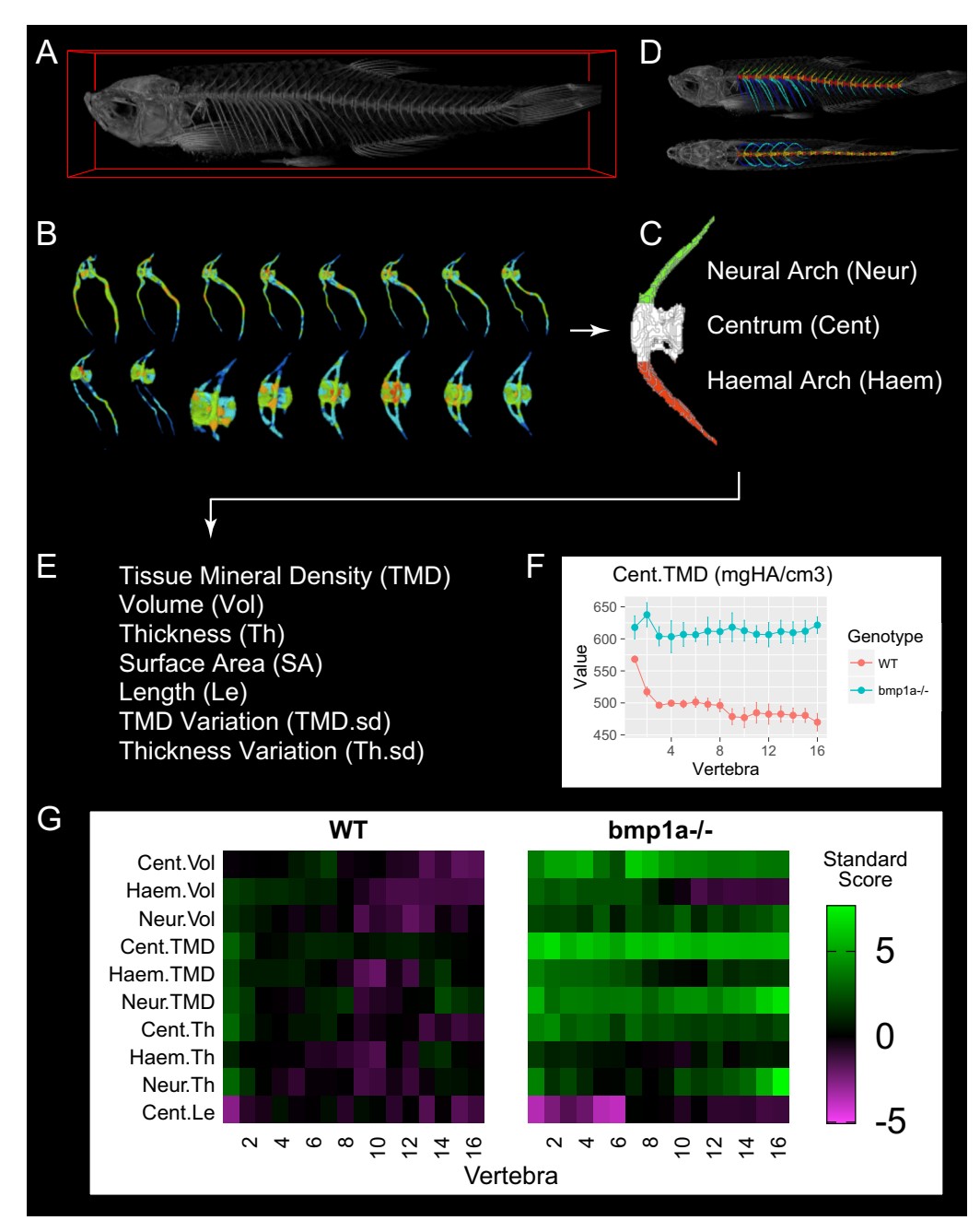

**Figure 1.** MicroCT-based phenomics in the axial skeleton of adult zebrafish. (**A**) MicroCT scans are acquired in adult fish (image shows 3D volume rendering). (**B**) For segmentation, individual vertebrae are isolated using FishCuT (image shows 16 vertebrae from the same fish, colors indicate local thickness), and (**C**) each vertebral body is segmented into three skeletal elements: the Neural Arch (green), Centrum (white), and Haemal Arch/Pleural Ribs (red). A representative segmentation can be seen in (**D**), which depicts the same fish in (**A**) with an alternating color scheme used to highlight individual skeletal elements segmented by FishCuT (top: lateral view; bottom: anteroposterior view). (**E**) For each skeletal element, FishCuT computes the following: Tissue Mineral Density, Volume, Thickness, Surface Area, Length (centrum only), Tissue Mineral Density Variation, and Thickness Variation. (**F**) For analysis, each combination of element/outcome is computed as a function of vertebra number, and subjected to the global test. Shown is a plot of one combination of element/outcomes, Cent.TMD, as a function of vertebra number in WT vs. *bmp1a*[-/-] fish. (**G**) Standard scores are computed and arranged into 'skeletal barcodes' that facilitate visualization of phenotypic trends both within individuals and across groups. Shown are the skeletal barcodes for a single WT (left) and *bmp1a*[-/-] fish (right).

DOI: https://doi.org/10.7554/eLife.26014.002

correlation coefficients for each pair of measurements across 34 analyzed fish. We observed a relatively low median absolute Pearson correlation coefficient of 0.34 among measures (*Figure 2*, *Figure 2—source data 1*), similar to the value of 0.3 attained by Pardo-Martin et al. during high-content profiling of the zebrafish craniofacial skeleton (*Pardo-Martin et al., 2013*). This quantity was relatively invariant with increasing number of vertebrae used for analysis (*Figure 2—figure supplement 1*). In the discussion, we provide examples (including fish analyzed in this study) where each vertebra might confer non-redundant information.

## MicroCT imaging at medium and high resolution results in comparable ability to distinguish mutant phenotypes in *bmp1a*<sup>-/-</sup> fish

Due to the large volumes that must be acquired, microCT scanning of the entire vertebral column in adult zebrafish requires a balance between image resolution and throughput. We assessed (a) the correlation in measurements quantified using scans performed at 21 µm (medium) vs. 10.5 µm (high) nominal isotropic resolution, and (b) the sensitivity of these two resolutions in discriminating mutant phenotypes in known skeletal mutants (as is customary we consider nominal isotropic resolution to be equivalent to isotropic voxel size; please see (*Bouxsein et al., 2010*) for detailed definitions). For testing, we used *bmp1a*<sup>-/-</sup> mutant fish. In humans, mutations in *BMP1* result in a rare recessive form of the brittle bone disease Osteogenesis Imperfecta (OI) (*Asharani et al., 2012*). In zebrafish, the *frilly fins* (*frf*) mutant harbors mutations in *bmp1a* (*Asharani et al., 2012*) and exhibits high vertebral bone mineral density (BMD) in adults, mimicking the high BMD phenotype in humans with OI caused by *BMP1* mutations (*Asharani et al., 2012*). We serially scanned n = 3 *bmp1a*<sup>-/-</sup> mutant fish and WT sibling controls (SL of WT: 25.7 ± 1.2 mm, SL of mutant: 23.2 ± 1.0 mm, mean ±SD, SL = standard length) at medium and high resolutions, analyzed scans using FishCuT, and compared quantities from individual vertebrae via linear regressions (*Figure 3*, *Figure 3—source data 1*). In general, we observed extremely high correlations between values attained at the two scanning resolutions, with $R^2$ values ranging from 0.98 to >0.99. In most cases, slopes deviated slightly from unity: slopes ranged from 0.90 to 0.91 for TMD measurements, while it ranged 1.18 to 1.21 for thickness measurements (we did not observe a consistent trend for volumetric measurements). This deviation from unity is likely to due to partial volume effects (PVEs; an inherent property of microCT images that emerges from projecting a continuous object onto a discrete grid [*Rittweger et al., 2004*]). However, due to the high correlation in measurements attained at each resolution, we observed minimal impact of PVEs on sensitivity in discriminating mutant phenotypes, as t-tests for differences in single vertebrae between WT and *bmp1a*<sup>-/-</sup> fish yielded similar p-values regardless of scan resolution used (see *Figure 3* for values). These findings were reproducible in multiple vertebrae (*Figure 3—figure supplement 1*); a comprehensive summary of findings across all phenotypic measurements is provided in *Figure 3—figure supplement 2*. Collectively, these analyses suggest that microCT scans at high and medium resolutions provided highly correlated information on each measure, and were comparable in their ability to discriminate WT from mutant phenotypes.

## Phenomic profiling enhances sensitivity in discriminating mutant populations

As described above, for statistical testing we developed a procedure whereby each combination of element/outcome is computed as a function of vertebra number, and the global test is used to assess whether the pattern across vertebrae is significantly different between groups. We hypothesized that assessing vertebral patterns with the global test would provide greater sensitivity in distinguishing mutant populations compared to (a) t-tests of individual vertebrae, and (b) t-tests of quantities averaged across all vertebrae. Note that for the rest of the manuscript, we restrict our analysis to the 16 anterior-most vertebrae; this set of vertebrae contains both high (~8) and low (~8) haemal arch volumes (e.g., see Figure 5F). For each combination of element/trait (e.g., Cent.TMD), we computed the power of the global test when assessing this measure in vertebrae 1–16, and compared it to t-tests of this measure in vertebra 2, or t-tests of this measure when averaged across vertebrae 1–16.

To test our hypothesis, we performed Monte Carlo simulations (*Figure 4—source data 1*). We first examined the potential to model the pattern of each combination of skeletal element/trait (e.g., Cent.TMD) for k = 16 vertebrae using a multivariate normal distribution. We analyzed n = 16 WT fish

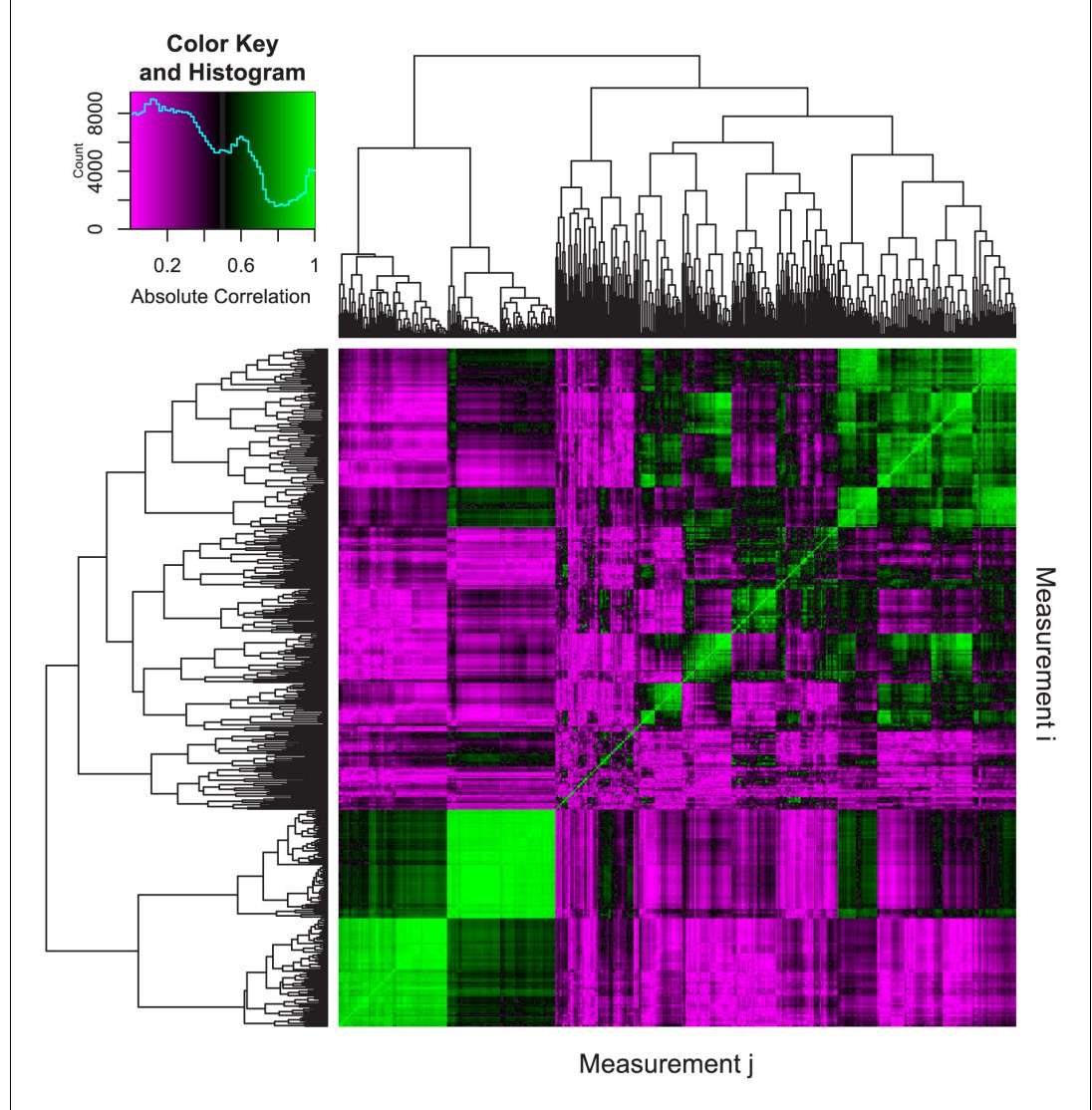

**Figure 2.** Pairwise correlations of phenotypic measurements. The following procedure was used to assess the correlation between phenotypic measurements. First, we analyzed 34 different WT and mutant fish of various genotypes. As detailed in the text, analyzing 24 vertebrae results in 24 × 25 = 600 phenotypic measures for each fish. We denote measure i in fish n as $x_i^n$ (i = 1 to 600 and n = 1 to 34). Next, we computed absolute Pearson correlation coefficients between each pair of measurements as $|\rho_{i,j}|=|cov(X_i,X_j)/(\sigma(X_i)\sigma(X_j))|$ where $X_i=[x_i^1, x_i^2,...,x_i^{34}]$, $X_j=[x_j^1, x_j^2,...,x_j^{34}]$, cov is the covariance, and σ is the standard deviation. The resulting 600 × 600 correlation matrix was plotted as a heatmap, where the element in the ith row and jth column represents the absolute Pearson correlation between measurement i ($X_i$) and measurement j ($X_j$). For each pairwise correlation, if a fish had missing values (e.g., not all 24 vertebrae were analyzed), it was excluded from the analysis. Hierarchical clustering was used to order the measurements in order to facilitate visualization. The predominance of purple regions indicates the prevalence of measurement pairs that exhibit low correlation. The legend depicts a color scale and a histogram of absolute correlations (computed from 600 × 600 = 360,000 different pairwise correlations). The median value of this distribution was 0.34.

DOI: https://doi.org/10.7554/eLife.26014.003

The following source data and figure supplement are available for figure 2:

**Source data 1.** Zip file containing phenotypic data (one text file per fish) as well as R code used for analysis.

DOI: https://doi.org/10.7554/eLife.26014.005

**Figure supplement 1.** Median absolute correlation plotted as a function of number of vertebrae analyzed.

DOI: https://doi.org/10.7554/eLife.26014.004

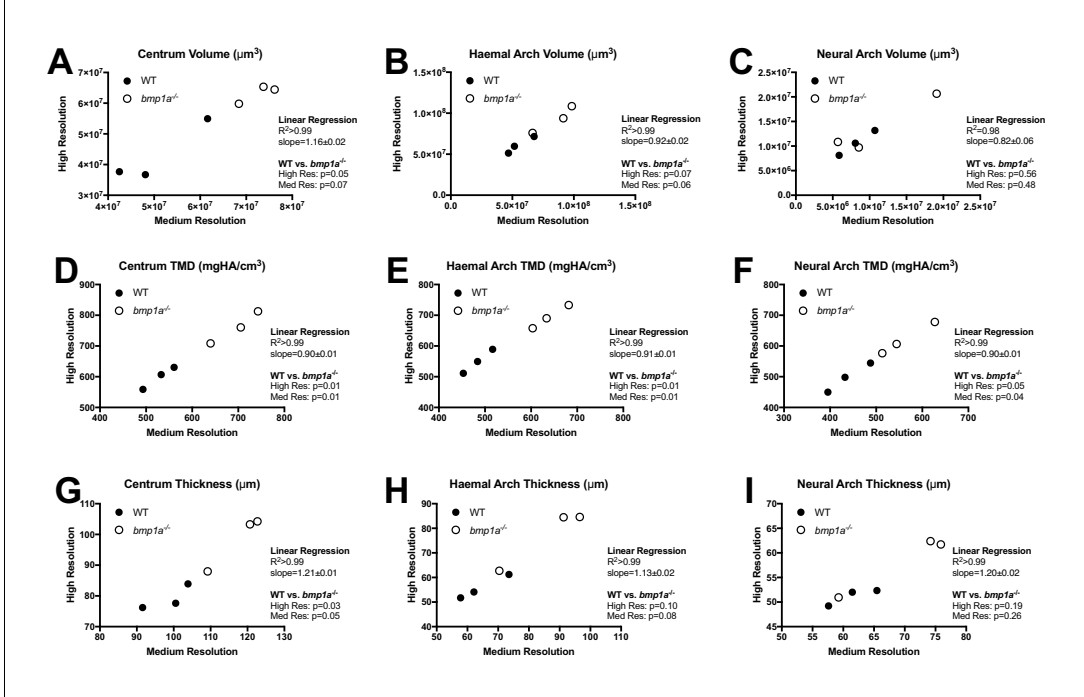

**Figure 3.** Analysis of WT and *bmp1a*[-/-] mutant fish reveals high correlation in phenotypic measurements quantified via medium and high resolution microCT. Results are shown for the second pre-caudal vertebrae for nine different measurements (A–I). Each point represents a single animal, with the corresponding measurement assessed at two nominal isotropic resolutions: medium (21 μm voxel size) and high (10.5 μm voxel size). WT animals (n = 3) are depicted as closed circles and *bmp1a*[-/-] fish (n = 3) as open circles. Linear regressions revealed a high level of correlation. Further, t-tests showed similar p-values when comparing WT vs *bmp1a*[-/-] at each resolution.
DOI: https://doi.org/10.7554/eLife.26014.006

The following source data and figure supplements are available for figure 3:

**Source data 1.** Zip file containing phenotypic data (one text file per fish) as well as R code used for analysis.
DOI: https://doi.org/10.7554/eLife.26014.009

**Figure supplement 1.** Phenotypic measurements quantified via medium and high resolution microCT scans for the first (A–I) and third (A'–I') pre-caudal vertebrae.
DOI: https://doi.org/10.7554/eLife.26014.007

**Figure supplement 2.** Heatmaps summarizing results from linear regressions in *Figure 3* (Vert 2) and *Figure 3—figure supplement 1* (Verts 1 and 3), with all phenotypic measurements reported.
DOI: https://doi.org/10.7554/eLife.26014.008

(SL: 20.4 ± 0.9 mm) from the same clutch, and performed Royston tests for multivariate normality for each combination of skeletal element/trait. For these studies we limited our analyses to subsets of k = 8 vertebrae (Group A: Vert 1, 3, 5, 7, 9, 11, 13, and 15; Group B: and Vert 2, 4, 6, 8, 10, 12, 14, and 16) so that the number of variables was less than the sample size (*Figure 4—source data 2*). The majority of phenotypic features (21 out of 25) were associated with non-significant p-values in both groups, suggesting these data exhibited multivariate normality. The four phenotypic features associated with significant p-values in either group – Cent.Th, Neur.Th, Haem.Th.sd, and Cent.Le – were excluded from the rest of our Monte Carlo simulations. Next, for each combination of skeletal element/trait we computed means (denoted by $\mu_i^{WT}$, where $\mu_i$ is the mean in vertebra i) and covariances (denoted by $\sum_{ij}^{WT}$, where $\sum_{ij}$ is the covariance between vertebra i and vertebra j) for k = 16 vertebrae. These parameter estimates were used to construct multivariate normal distributions for Monte Carlo simulations (≥10,000 simulations per analysis) (*Figure 4*). For each analysis, we constructed two multivariate normal distributions: (1) a WT distribution using means $\mu_i^{WT}$, and (2) a 'simulated' mutant distribution using means $\mu_i^{WT} + d*\sigma_i^{WT}$ (where d is a characteristic effect size, and $\sigma_i$ is the standard deviation in vertebra i). Covariances were assumed to be equal in both distributions, and set to $\sum_{ij}^{WT}$. We first estimated the power of the global test in discriminating high Tot.TMD in simulated mutant fish as a function of sample size and alpha value (*Figure 4A*). For these

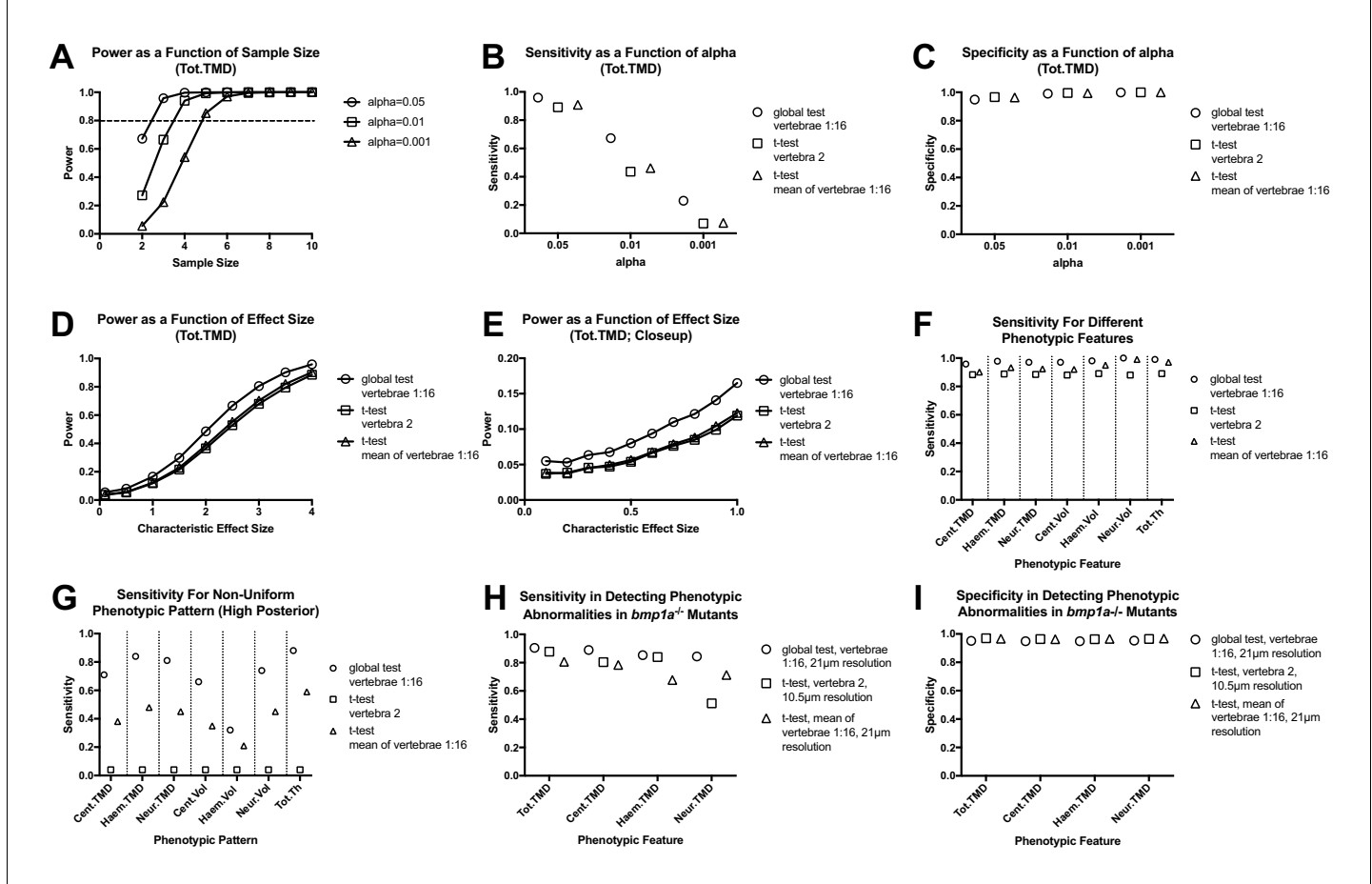

**Figure 4.** Analyzing vertebral patterns confers heightened sensitivity in discriminating mutant phenotypes compared to analyzing individual vertebrae. Figures A-G show results from Monte Carlo simulations using simulated mutant phenotypes. (A) Power of the global test in discriminating a uniform increase in Tot.TMD (characteristic effect size: d = 4) as a function of sample size and alpha value. The dotted line highlights sample sizes with power >0.8. (B) Sensitivity (fraction of times in which p<0.05 when comparing simulated mutant vs. WT fish) for different test procedures. (C) Specificity (1 - fraction of times in which p<0.05 when comparing WT vs. WT fish) for different test procedures. (D) Power as a function effect size for different test procedures (sample size: n = 3; alpha = 0.05). (E) Closeup of figure (D) for smaller effect sizes. (F) Comparison of sensitivity for different phenotypic features (characteristic effect size: d = 4; sample size: n = 3; alpha = 0.05). (G) Sensitivity for non-uniform phenotypic pattern (linear increase from d = 0 at vert 1 to d = 4 at vert 16; n = 3; alpha = 0.05). Figure H-I show results from Monte Carlo simulations using parameter estimates derived from *bmp1a*[-/-] mutants. (H) Sensitivity in discriminating different phenotypic features in *bmp1a*[-/-] mutants. T-tests using individual vertebrae were performed on data derived from fish scanned at high resolution. (I) Specificity in discriminating different phenotypic features in *bmp1a*[-/-] mutants.

DOI: https://doi.org/10.7554/eLife.26014.010

The following source data is available for figure 4:

**Source data 1.** Zip file containing phenotypic data (one text file per fish) as well as R code used for analysis.
DOI: https://doi.org/10.7554/eLife.26014.011
**Source data 2.** Summary of Royston test results.
DOI: https://doi.org/10.7554/eLife.26014.012
**Source data 3.** Summary of sensitivity and specificity for different test procedures.
DOI: https://doi.org/10.7554/eLife.26014.013

simulations, we assumed a characteristic effect size of d = 4, since in the *bmp1a*[-/-] mutants from the previous section, Tot.TMD was increased ~4 standard deviations above the mean in each vertebra. We found that a power of >0.8 was attained with a sample size of n = 3, 4, and 5 fish/group for alpha values of 0.05, 0.01, and 0.001, respectively. Next, using a sample size of n = 3 fish/group and alpha = 0.05, we estimated test sensitivity (fraction of times in which p<0.05 when comparing simulated mutant fish to WT fish) (*Figure 4B*) and specificity (1 - fraction of times in which p<0.05 when

comparing WT to WT fish) (*Figure 4C*) for (a) the global test (using vertebrae 1:16), (b) t-tests of averaged quantities (using the mean of vertebrae 1:16), and (c) t-tests of individual vertebrae (using vertebra 2). We found that the global test conferred higher sensitivity, with similar specificity, compared to t-tests of averaged quantities as well as t-tests of individual vertebrae. For instance, at alpha = 0.05, the sensitivity of the global test, t-test of individual vertebrae, and t-test of averaged quantities was 0.96, 0.89, and 0.91. At alpha = 0.01, the sensitivity of the global test, t-test of individual vertebrae, and t-test of averaged quantities was 0.67, 0.43, and 0.46 (respectively). Specificity values were consistent with those predicted from specified levels of alpha: as an example, at alpha = 0.01, specificity was 0.99 in all three cases. For alpha = 0.05 and n = 3, we estimated a false discovery rate of 5.1% for the global test procedure.

Next, we compared the power of the different testing procedures as a function of characteristic effect size (*Figure 4D,E*). We found that the global test conferred higher power compared to t-tests of averaged quantities and t-tests of individual vertebrae, with the relative increase greatest at small characteristic effect sizes (e.g.,~1.4 to 1.5 fold greater at an effect size of 0.5, compared to ~1.2 fold greater at an effect size of 3). Higher sensitivity in the global test was also observed when we extended our analysis to other phenotypic features (*Figure 4F*, *Figure 4—source data 3*).

Finally, we examined the effects of non-uniform phenotypic abnormalities on test sensitivity. Specifically, we constructed a new simulated mutant distribution using means $\mu_i^{WT} + d_i * \sigma_i^{WT}$, where $d_i = [(i-1)/(k-1)]*d$ is the characteristic effect size at vertebra i, and k is the total number of vertebrae analyzed. We set d = 4 and k = 16 such that the characteristic effect size at each vertebra linearly varies from $d_1 = 0$ to $d_{16} = 4$. Using this 'high anterior' pattern, the differences in test sensitivity became magnified (*Figure 4G*). For instance, for Cent.TMD, sensitivity of the global test was ~2 fold that of t-tests of averaged quantities, and ~18 fold that of t-tests of individual vertebra. Notably, our studies also suggested that power in the global test is affected by the assumed phenotypic pattern. For example, for Cent.TMD, sensitivity for the global test using a non-uniform pattern was 0.71, compared to 0.96 for a uniform pattern (as in *Figure 4F*).

We sought to corroborate the above findings using an independent experimental cohort. Specifically, we used the n = 3 *bmp1a*$^{-/-}$ and sibling WT controls (*bmp1a*$^{+/+}$) from the previous section to construct two multivariate normal distributions: 1) a WT distribution using means $\mu_i^{bmp1a+/+}$ and covariances $\sum_{ij}^{bmp1a+/+}$, and 2) a mutant distribution using means $\mu_i^{bmp1a-/-}$ and covariances $\sum_{ij}^{bmp1a-/-}$, and repeated Monte Carlo simulations (*Figure 4H–I*). Note that these sibling controls are identified as *bmp1a*$^{+/+}$ to distinguish them from the WT fish used for the Monte Carlo simulations in *Figure 4A–G*. Consistent with our analyses using simulated mutants, we found that when using parameter estimates from *bmp1a*$^{-/-}$ mutants, the global test conferred higher sensitivity, with similar specificity, compared to t-tests of averaged quantities as well as t-tests of individual vertebrae, even when the latter was performed at higher resolution. For instance, for Cent.TMD (and assuming n = 3 and alpha = 0.05), the sensitivity of the global test, t-test of individual vertebrae, and t-test of averaged quantities was 0.89, 0.80, and 0.78, and specificity was 0.95, 0.96, and 0.96. Similar results were observed for Tot.TMD, Haem.TMD, and Neur.TMD.

## Identification of novel phenotypic features in known axial skeletal mutants

In employing zebrafish as a predictive model of human skeletal gene function it is important to understand how zebrafish mutant skeletal phenotypes relate to orthologous mutations associated with human genetic bone disorders. To identify multivariate zebrafish phenotypes associated with human brittle bone disease, we performed comprehensive phenotypic characterization of *bmp1a*$^{-/-}$ fish, as well as another mutant associated with this disorder, *plod2*$^{-/-}$ (*Gistelinck et al., 2016a*). Mutations in *PLOD2* are associated with Bruck syndrome, a recessive condition resembling OI. Previous studies revealed high vertebral BMD in adult zebrafish with mutants in *bmp1a*, (*Asharani et al., 2012*), however, it is unclear whether this high BMD is attributable to an increase in bone mass (e.g., increase in bone volume), and/or mineralization (e.g., increase in tissue mineral density). For *plod2*$^{-/-}$ mutants, adult animals were found to exhibit vertebral compressions, distorted vertebrae, and excessive bone formation (*Gistelinck et al., 2016a*). Previous microCT analyses showed higher Centrum TMD in these mutants. However, due to the lack of robust methods to analyze the highly dysmorphic vertebrae in *plod2*$^{-/-}$ fish, previous microCT analyses were performed in two dimensional

maximum intensity projections (*Gistelinck et al., 2016a*), and an in-depth, three dimensional phenotypic characterization of *plod2⁻/⁻* fish has yet to be performed.

Using FishCuT, we analyzed microCT scans of *bmp1a⁻/⁻* and *plod2⁻/⁻* mutant fish, and compared them to WT sibling controls (n = 3/group) (*Figure 5*, *Figure 5—source data 1*). Note that for the rest of our studies, we focus our analyses on the results of ten features (the nine possible combinations of (Cent, HA, NA) x (Vol, TMD, and Th), plus Cent.Le). In *bmp1a⁻/⁻* mutants, analysis revealed significant increases in both bone mass and mineralization. Specifically, in regard to bone mass, centrum volume was significantly increased (Cent.Vol: p=0.007). Further, bone thickness was significantly elevated in all skeletal elements (Cent.Th: p=0.0007, Haem.Th: p=0.048, Neur.Th: p=0.005). No significant differences were observed with respect to haemal/neural arch volume (Haem.Vol:

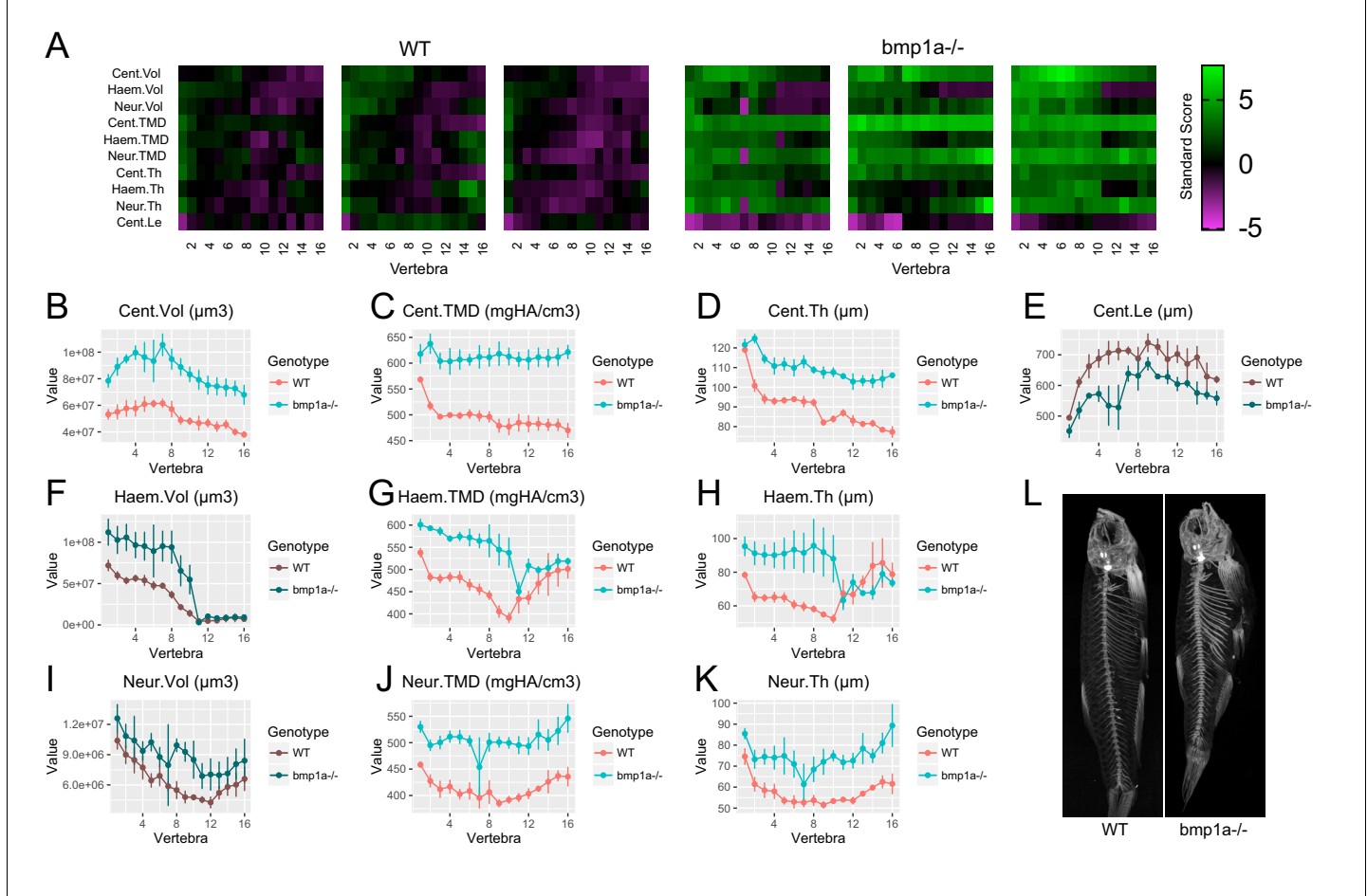

**Figure 5.** Analysis of *bmp1a⁻/⁻* fish. (**A**) Skeletal barcodes for WT and *bmp1a⁻/⁻* fish (n = 3/group). Each barcode represents a single fish. Standard scores are computed as the difference between the value of the feature in the individual and the mean value of the feature across all vertebrae in the control population, divided by the standard deviation of the feature across all vertebrae in the control population (see text for details). (**B–K**) Phenotypic features plotted as a function of vertebra (mean ± SE, n = 3/group). Plots associated with a significant difference are colored in a lighter coloring scheme (see text for p-values). The same plots with y axis set to zero are shown in *Figure 5—figure supplement 2*. (**L**) Maximum intensity projection of microCT scans.

DOI: https://doi.org/10.7554/eLife.26014.014

The following source data and figure supplements are available for figure 5:

**Source data 1.** Zip file containing phenotypic data (one text file per fish) as well as R code used for analysis.
DOI: https://doi.org/10.7554/eLife.26014.017
**Figure supplement 1.** Covariate analysis of Haem.TMD in *bmp1a⁻/⁻* fish.
DOI: https://doi.org/10.7554/eLife.26014.015
**Figure supplement 2.** Same data as in *Figure 5B–5K* with y axes set to zero.
DOI: https://doi.org/10.7554/eLife.26014.016

p=0.07, Neur.Vol: p=0.16) or centrum length (Cent.Le: p=0.051). In regard to bone mineralization, TMD was significantly elevated in all skeletal elements (Cent.TMD: p=0.003, Haem.TMD: p=0.006, Neur.TMD: p=0.006). While Cent.TMD and Neur.TMD appeared to be uniformly elevated across all vertebrae, Haem.TMD appeared to be differentially elevated in precaudal (first ten) vertebrae. We used the *covariates* functionality in the *globaltest* package in *R* to identify vertebral clusters that exhibit a significant association with genotype. We found that only vertebrae 2, 3, 4, and 7 were significantly associated with genotype (*Figure 5—figure supplement 1*), consistent with the notion that precaudal vertebrae were differentially affected.

For *plod2*[−/−] mutants (SL of WT: 27.9 ± 0.7 mm, SL of mutant: 20.2 ± 1.1 mm, mean ±SD) (*Figure 6*, *Figure 6—source data 1*), we found that FishCuT robustly segmented vertebrae, despite severe vertebral malformations. Consistent with previous studies, we observed significantly elevated TMD in the centrum (Cent.TMD: p=0.038) but not haemal/neural arches (Haem.TMD: p=0.91, Neur.TMD: p=0.53) (*Gistelinck et al., 2016a*). Further, we observed a significant decrease in centrum length

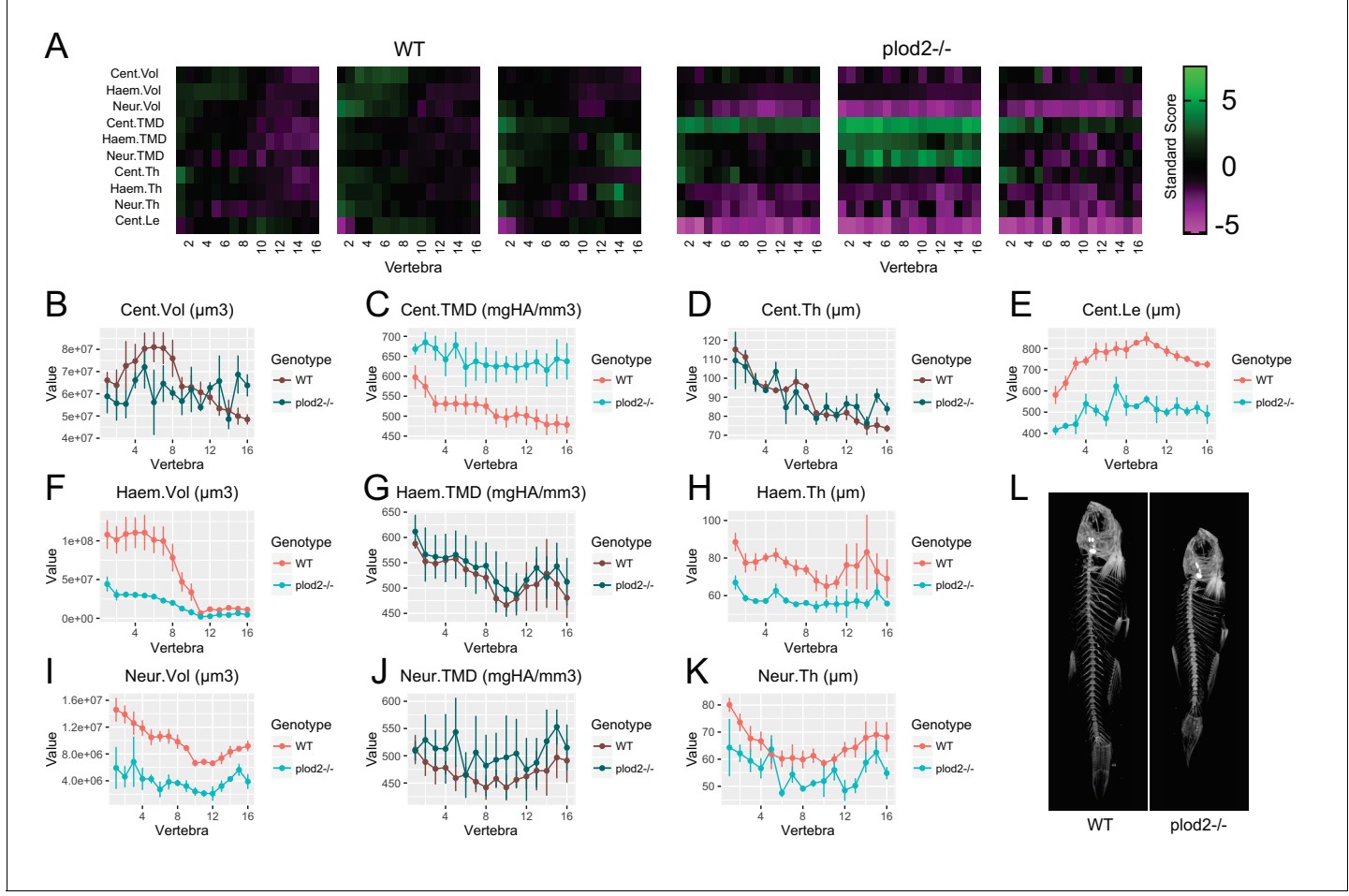

**Figure 6.** Analysis of *plod2*[−/−] fish. (**A**) Skeletal barcodes for WT and *plod2*[−/−] fish (n = 3/group). WT fish in this figure are different from those in *Figure 5*. (**B–K**) Phenotypic features plotted as a function of vertebra (mean ±SE, n = 3/group). Plots associated with a significant difference are colored in a lighter coloring scheme (see text for p-values). The same plots with y axis set to zero are shown in *Figure 6—figure supplement 1*. (**L**) Maximum intensity projection of microCT scans.

DOI: https://doi.org/10.7554/eLife.26014.018

The following source data and figure supplement are available for figure 6:

**Source data 1.** Zip file containing phenotypic data (one text file per fish) as well as R code used for analysis.
DOI: https://doi.org/10.7554/eLife.26014.020

**Figure supplement 1.** Same data as in *Figure 6B–6K* with y axes set to zero.
DOI: https://doi.org/10.7554/eLife.26014.019

(Cent.Le: p=6.5×10$^{-5}$), as reported previously (*Gistelinck et al., 2016a*). Morphologically, *plod2*$^{-/-}$ mutants exhibited a low bone mass phenotype in the haemal and neural arches (Haem.Vol: p=0.018, Neur.Vol: p=0.012), but not centrum (p=0.15). Further, *plod2*$^{-/-}$ mutants exhibited decreased haemal and neural arch thickness (Haem.Th: p=0.013, Neur.Th: p=0.034), but not centrum thickness (p=0.47).

## Allometric models aid in discriminating mutant phenotypes masked by alterations in growth

In zebrafish, developmental progress is more closely related to standard length than to age (*Parichy et al., 2009*; *McMenamin et al., 2016*). In analyzing mutants that exhibit differences in body size (e.g., *plod2*$^{-/-}$ mutants exhibited severely diminished body size compared to WT siblings), it is difficult to discriminate to what degree altered phenotypes are attributable to differences in developmental progress, versus specific effects on skeletal function. Furthermore, although morphological developmental milestones can sometimes allow staging despite genotype-specific differences in size and growth during the larval-to-adult transformation, few such milestones have been identified, particularly during later stages. In addition, some milestones are themselves skeletal traits, necessitating an alternative approach. To help address these challenges, we developed allometric models to control for effects of body size during phenomic analysis. To model each phenotypic feature in WT fish as a function of standard length, we used a standard power-law relationship for allometric modeling (*Lleonart et al., 2000*):

$$y = ax^b \quad (1)$$

where *y* is the feature of interest, *x* is standard length, and *a* and *b* are empirically-derived parameters. The scaling exponent *b* is directly interpretable when quantities are associated with mass, length, area, and volume. Thus, we converted TMD to a mass-based quantity by computing tissue mineral content (TMC, mgHA) as the product of volume and TMD in each skeletal element (e.g., Cent.TMC = Cent.Vol*Cent.TMD). To attain estimates of a and b for WT animals (*Figure 7—source data 1*), we performed an ontogenetic series by profiling n = 16 WT fish over a range of standard lengths (18.4 mm to 31.8 mm), and fit these data to the power-law relationship to estimate a and b (*Figure 7—figure supplement 1*). Convergence analyses demonstrated that model parameters were relatively invariant when more than ~10 samples were included in the analysis (*Fig 7—figure supplement 2*), suggesting that our use of n = 16 samples was sufficient to provide reliable model parameter estimates. In general, we found that most features significantly deviated from isometric growth (i.e., proportional relationships were not preserved with growth) with respect to standard length (*Figure 7—source data 2*). Specifically, features associated with thickness and volume exhibited negative allometry (scaling exponents lower than those expected for isometric growth), while TMC exhibited positive allometry.

Next, we used the following relationship to normalize for allometric effects of growth (*Lleonart et al., 2000*):

$$y^* = y(x^*/x)^b \quad (2)$$

where y is the feature of interest, x is standard length, x* is a reference standard length, and y* is the transformed value. It is important to point out that once estimates of the scaling exponent b is obtained in a comprehensive sample (as in in the n = 16 fish above), this relationship can be applied to other WT fish in experimental groups of an arbitrary sample size. When we applied this relationship to the phenomic profiles from the WT animals in our ontogenetic series (using the mean standard length of all WT fish as the reference standard length), we found that the coefficient of variability was substantially reduced compared to unnormalized values, as well as when quantities were normalized by an alternate transformation, 1/SL (*Figure 7*).

Next, we used the above normalization procedure to re-analyze *plod2*$^{-/-}$ mutants. We scaled phenotypic data in WT sibling controls by applying *Equation 2*, using the mean standard length of *plod2*$^{-/-}$ mutants as the reference length. We did not scale phenotypic data in *plod2*$^{-/-}$ mutants, as allometric scaling is likely to be different than in WT animals. In our unnormalized analysis of *plod2*$^{-/-}$ fish (i.e., *Figure 6*), we observed significant decreases in several morphological quantities including Haem.Vol, Haem.Th, Neur.Vol, Neur.Th, and Cent.Le. In contrast to these low bone mass

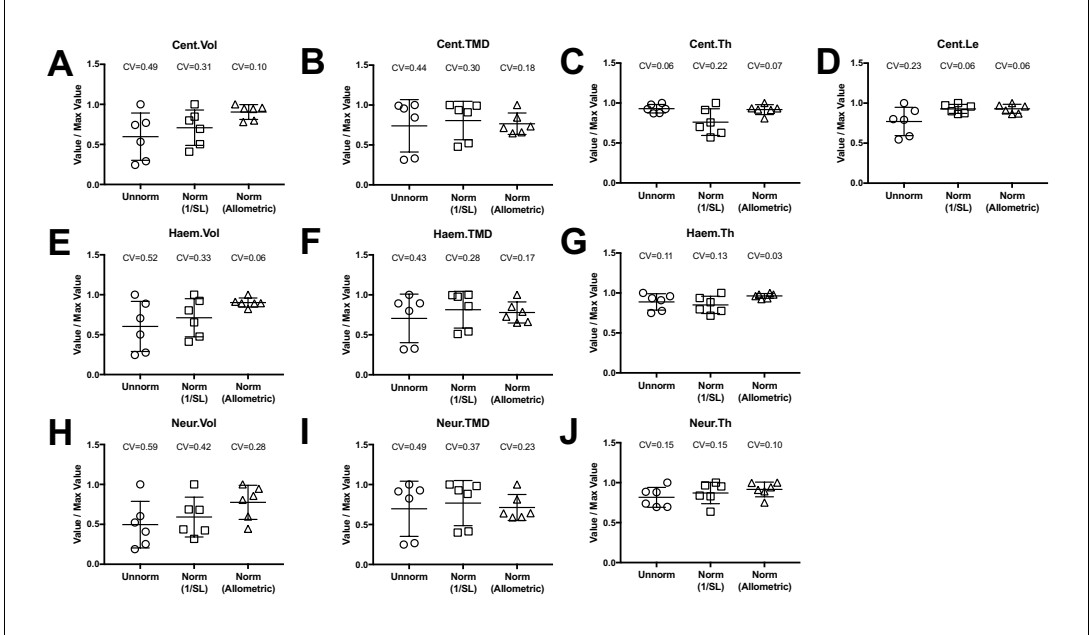

**Figure 7.** Allometric normalization for differences in body size. Data are shown for the first precaudal vertebra of n = 6 WT fish of different standard lengths. The coefficient of variability (CV) is shown for each normalization procedure. The left column shows high phenotypic variability in unnormalized data. When data were normalized using allometric models (right column), phenotypic variability was substantially reduced. Phenotypic variability was also reduced, though to a lesser extent, when data were normalized using an alternate normalization, 1/SL (middle column).

DOI: https://doi.org/10.7554/eLife.26014.021

The following source data and figure supplements are available for figure 7:

**Source data 1.** Zip file containing phenotypic data (one text file per fish) as well as R code used for analysis.
DOI: https://doi.org/10.7554/eLife.26014.024
**Source data 2.** Summary of scaling exponents (i.e., values of b) computed in allometric models.
DOI: https://doi.org/10.7554/eLife.26014.025
**Figure supplement 1.** Allometric modeling of phenotypic features.
DOI: https://doi.org/10.7554/eLife.26014.022
**Figure supplement 2.** Convergence analysis for allometric modeling.
DOI: https://doi.org/10.7554/eLife.26014.023

phenotypes we did not observe any differences in these features following normalization (Haem.Vol: p=0.23, Haem.Th: p=0.33, Neur.Vol: p=0.58, Neur.Th: p=0.56, Cent.Le: p=0.39) (*Figure 8*, *Figure 8—source data 1*). Instead, we observed a significant *increase* in Cent.Vol (p=0.003). Further, we observed a significant increase in centrum, haemal arch, and neural arch TMD (Cent.TMD: p=0.0001, Haem.TMD: p=0.002, Neur.TMD: p=0.002). Since Cent.Le (p=0.39) and Cent.Th (p=0.08) were similar in *plod2*[-/-] fish and WT siblings, we surmised that the increase in Cent.Vol may be attributable to an increase in centrum diameter. Consistent with this idea, when we manually examined transverse sections in microCT images (*Figure 8L*), we observed a clear increase in centrum diameter in *plod2*[-/-] mutants relative to similarly-sized, non-sibling WT animals (and to a lesser extent, larger, sibling controls). This phenotype was not previously identified during the initial characterization of the *plod2*[-/-] mutant line (*Gistelinck et al., 2016b*).Collectively, these analyses identify a novel phenotypic feature, centrum expansion, in *plod2*[-/-] mutants, and suggest the utility of phenomic-based allometric models as a complimentary analytical tool to reveal mutant phenotypes masked by variations in growth.

### Identification of opallus *as a novel axial skeletal mutant*

Finally, based on the potential for FishCuT to identify novel phenotypes in known skeletal mutants, we examined the potential for FishCuT to identify novel axial skeletal mutants among fish populations derived from forward genetic screens. The zebrafish mutant *opallus* was derived from a

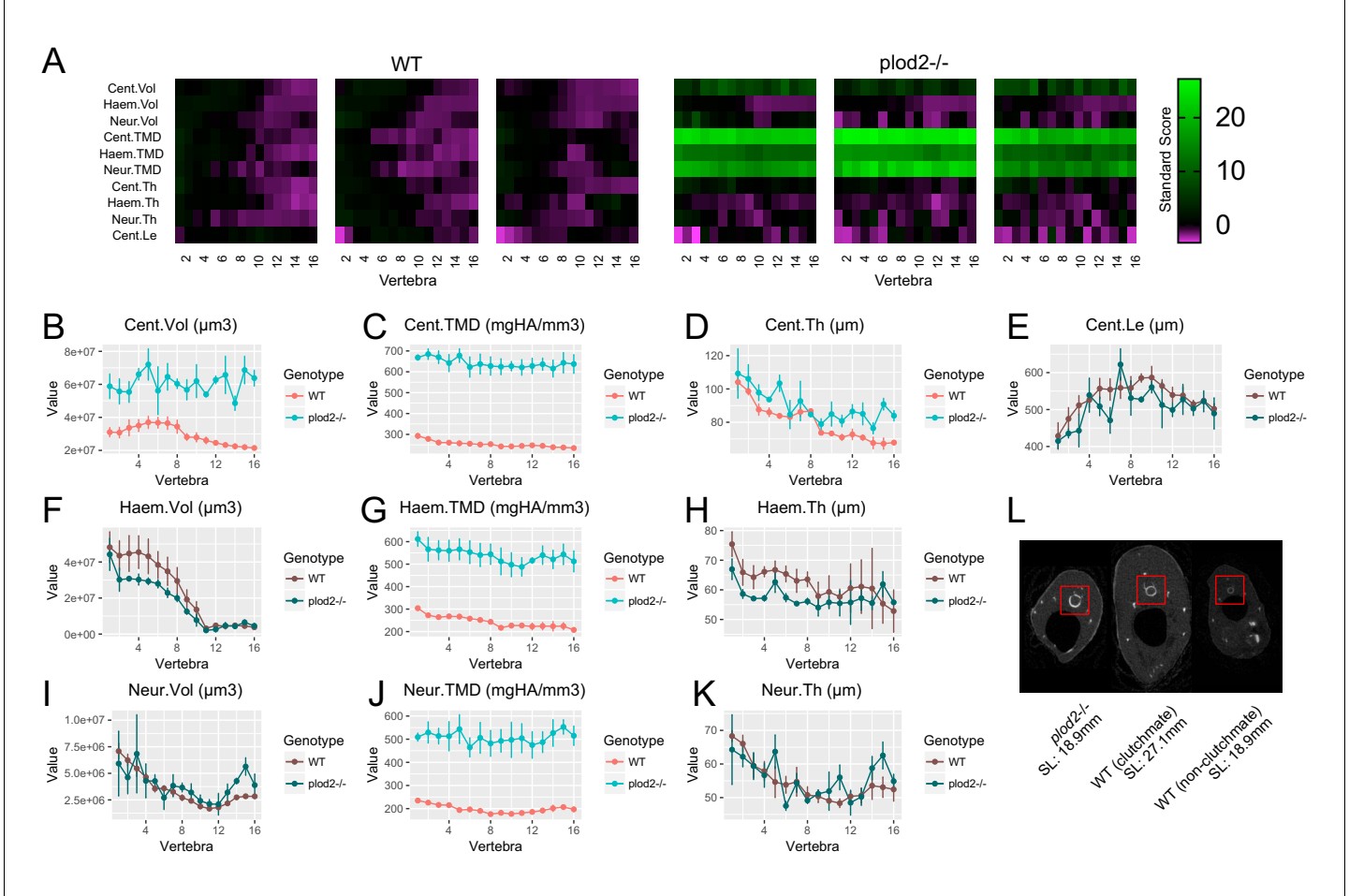

**Figure 8.** Analysis of *plod2⁻ᐟ⁻* fish following removal of allometric effects of body size. (**A**) Skeletal barcodes for WT and *plod2⁻ᐟ⁻* fish following removal of allometric effects of body size (n = 3/group). (**B–K**) Phenotypic features as a function of vertebra (mean ± SE, n = 3/group). Phenotypic data in WT sibling controls were subjected to allometric normalization; data in *plod2⁻ᐟ⁻* fish are identical to those in *Figure 6*. Plots associated with a significant difference are colored in a lighter coloring scheme (see text for p-values). Values for TMD were derived by a two-step process in which TMC and volume were subjected to allometric normalization independently, and normalized values for TMC and volume were used to calculate normalized values for TMD. The same plots with y axis set to zero are shown in *Figure 8—figure supplement 1*. (**L**) Transverse sections of microCT scans. Centra are highlighted by a red box in each animal. *plod2⁻ᐟ⁻* mutants (left) exhibit increased centrum diameter compared to standard length matched, non-clutchmate WT controls (right), and to a lesser extent, WT siblings (middle) of greater standard length. Images show posterior endplate of the sixth precaudal vertebra in all fish.

DOI: https://doi.org/10.7554/eLife.26014.026

The following source data and figure supplement are available for figure 8:

**Source data 1.** Zip file containing phenotypic data (one text file per fish) as well as R code used for analysis.
DOI: https://doi.org/10.7554/eLife.26014.028

**Figure supplement 1.** Same data as in *Figure 8B–8K* with y axes set to zero.
DOI: https://doi.org/10.7554/eLife.26014.027

forward genetic screen, and exhibits pigmentation abnormalities characterized by excessive xantho-phores and depleted melanophores, as well as jaw hypertrophy (*McMenamin et al., 2014*). *opallus* harbors a mutation in thyroid stimulating hormone receptor (*tshr*) identical to a human mutation causing constitutive TSHR activity and hyperthyroidism (*McMenamin et al., 2014*). These two conditions have been associated with opposing effects on human BMD. Specifically, while hyperthyroidism is traditionally associated with low BMD, TSHR gain-of-function has been associated with high BMD (*de Lloyd et al., 2010*); it was unknown whether *opallus* exhibited an axial skeletal phenotype. In a

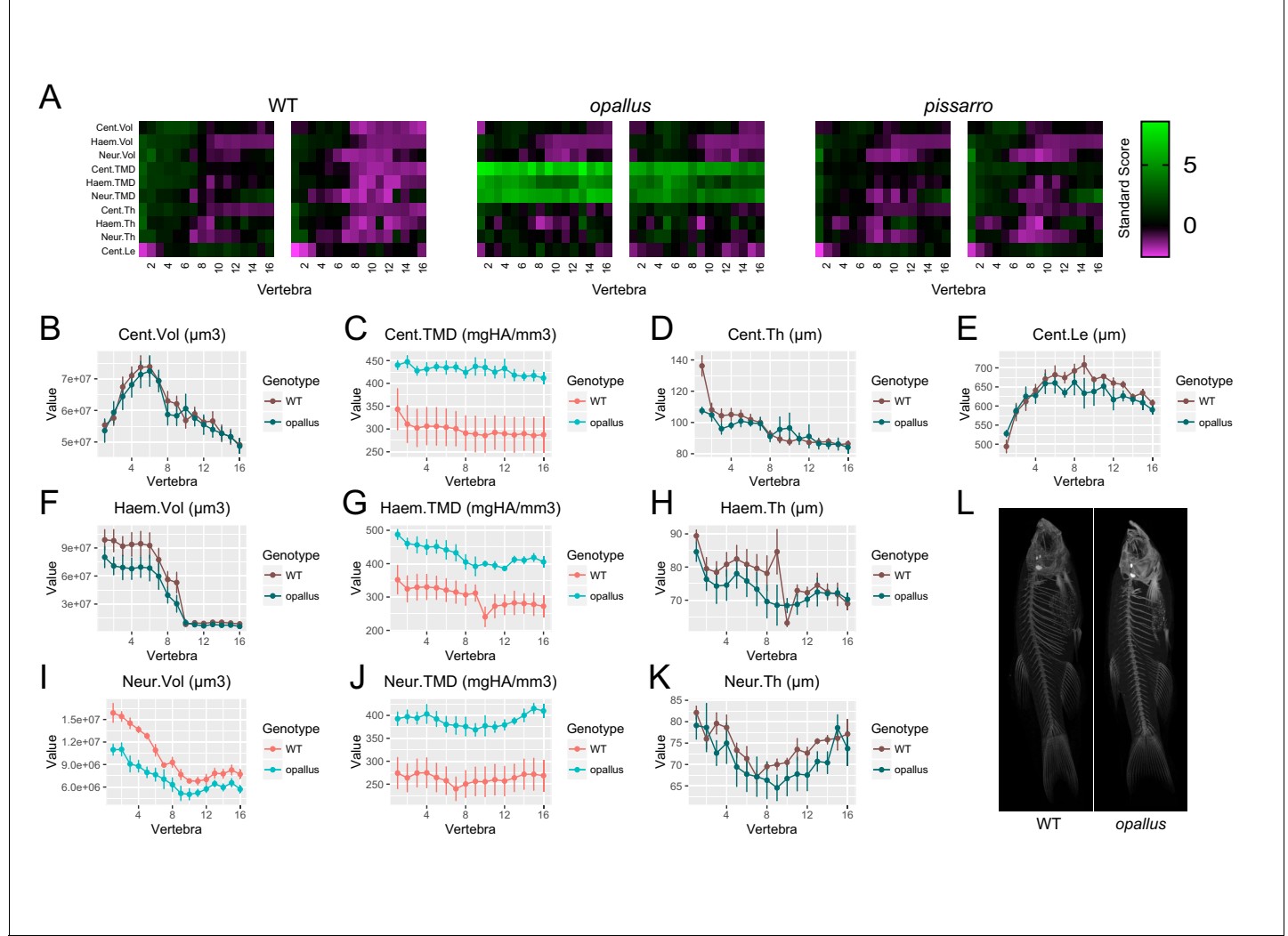

**Figure 9.** Identification of *opallus* as a novel axial skeletal mutant. (**A**) Skeletal barcodes for WT, *opallus*, and *pissarro* (n = 2 fish/group). Barcodes for *opallus*, but not *pissarro*, appear different from WT fish (**B–K**) Phenotypic features in *opallus* plotted as a function of vertebra (mean ±SE, n = 5/group). Plots associated with a significant difference are colored in a lighter coloring scheme (see text for p-values). The same plots with y axis set to zero are shown in *Figure 9—figure supplement 2*. (**L**) Maximum intensity projections of microCT scans.

DOI: https://doi.org/10.7554/eLife.26014.029

The following source data and figure supplements are available for figure 9:

**Source data 1.** Zip file containing phenotypic data (one text file per fish) as well as R code used for analysis.
DOI: https://doi.org/10.7554/eLife.26014.032
**Figure supplement 1.** Covariate analysis of Neur.Vol in *opallus*.
DOI: https://doi.org/10.7554/eLife.26014.030
**Figure supplement 2.** Same data as in *Figure 9B–K* with y axes set to zero.
DOI: https://doi.org/10.7554/eLife.26014.031

first cohort where we performed an initial screen of n = 2 fish (*Figure 9A*), we observed a clear increase in TMD in *opallus* that was not present in standard length matched AB controls, or in another mutant derived in a forward genetic screen that exhibits pigmentation defects, *pissarro* (*Quigley et al., 2004*) (SL of AB: 23.7 ± 1.1 mm, SL of *opallus*: 24.3 ± 1.4 mm, SL of *pissarro*: 23.7 ± 0.1 mm, mean ± SD). Follow up analyses (n = 5/group, SL of AB: 23.7 ± 0.5 mm, SL of *opallus*: 24.3 ± 1.4 mm, mean ± SD) revealed that *opallus* exhibited a significant increase in centrum, haemal arch, and neural arch TMD (Cent.TMD: p=0.012, Haem.TMD: p=0.009, Neur.TMD: p=0.006)

(*Figure 9B–K*, *Figure 9—source data 1*). Morphologically, most features were normal (Cent.Vol: p=0.88, Haem.Vol: p=0.20, Cent.Th: p=0.12, Haem.Th: p=0.38, Neur.Th: p=0.38, Cent.Le: p=0.29) except for neural arch volume, which was significantly decreased in *opallus* (Neur.Vol: p=0.002). This decrease was most pronounced in anterior vertebrae, with covariate analysis revealing significant associations between vertebrae 1–5 and genotype (*Figure 9—figure supplement 1*).

## Discussion

In this study, we developed microCT-based methods and a segmentation algorithm, FishCuT, enabling profiling of morphological and densitometric traits at a large number of anatomical sites in the axial skeleton of adult zebrafish. We profiled ~30,000 data points derived from ~3600 skeletal elements in wildtype fish of different degrees of developmental progress as well as mutant lines associated with human disease. Our studies reveal virtues of deep phenotyping in a single, complex organ system.

A challenge to the analysis of high-dimensional phenotypic data is the curse of dimensionality: when testing for changes in each feature individually, as a consequence of multiple testing correction, the number of samples required for a statistically reliable result increases exponentially with the number of features assessed. In lieu of analyzing each feature in isolation, we examined whether patterns of element/outcome combinations varied across vertebral bodies were altered in mutant populations. In Monte Carlo simulations, when holding alpha constant, power in the global test was consistently higher compared to t-tests of averaged quantities and t-tests of individual vertebrae, with the relative increase in power greatest at small effect sizes. Further, simulated levels of specificity were consistent with specified values of alpha. Our studies suggest that vertebral phenomic patterns may confer enhanced sensitivity in discriminating mutant phenotypes relative to analyzing individual vertebrae. This attribute may increase productivity in genetic screens, as well as provide opportunities to study genetic variants of smaller effect size, such as those which underlie the overwhelming majority of complex diseases (*Gibson, 2012*).

It is important to note that while our workflow is highly sensitive in discriminating mutants that exhibit subtle phenotypic alterations in a large number of bones, other scan resolutions and statistical testing may be appropriate in some cases. For instance, while we found that scans acquired at 21 μm and 10.5 μm resolution were comparable in their analysis of *bmp1a*[-/-] mutants, it is possible that other mutant lines may present extremely small, thin, or hypomineralized structures that require greater scanning resolution to resolve. In this context, while increasing nominal resolution did not increase power in detecting mutant phenotypes in *bmp1a*[-/-] mutants, these findings are not generalizable to all mutant phenotypes. Indeed, if changes in mutants are of a feature size much smaller than the scan resolution, increasing scan resolution would very likely increase assay sensitivity. Further, given the fact that the global test exhibits optimal power when many features exhibit minor changes (*Goeman et al., 2006*), our workflow will be most optimal when there are many small changes in a large number of vertebrae, while other statistical tests may be more powerful when testing for larger effects in a small number of vertebrae. Finally, the acquisition of additional high-resolution scans in select vertebrae may be desirable to safeguard against the possibility of missing phenotypic abnormalities that could escape detection at lower resolution.

Our studies make at least two contributions toward enabling rapid throughput analysis across a variety of different mutants/backgrounds – (1) developing the ability to analyze microCT images in less than 5 min/fish using FishCuT, and (2) demonstrating proof-of-concept of the ability to acquire whole-spinal microCT scans at 5 min/fish, if eight fish are scanned simultaneously. Nonetheless, several practical issues must be considered before broad application. This includes designing specimen holders with optimal animal packing (e.g., scanning eight fish simultaneously requires the fish to be physically touching, preventing automatic segmentation). In addition, potential imaging artifacts associated with highly multiplexed scanning strategies need to be characterized. For instance, when scanning two fish at a time, we are able to position each fish equidistant from the scan center. This minimizes potential for erroneous measurements due to beam hardening artifacts in the radial direction. When scanning eight fish at a time this is not possible, and thus effects from beam hardening artifacts could be more significant. Finally, it should be noted that analysis of microCT scans using FishCuT typically takes less than 5 min per animal; this throughput is most likely to be useful for

reverse genetic screens requiring analysis of hundreds of specimens, rather than extremely large forward genetic screens where ~ 10,000 animals may require analysis.

By phenotyping $bmp1a^{-/-}$ and $plod2^{-/-}$ mutants, our studies shed new light on multivariate phenotypes in the zebrafish skeleton associated with human skeletal disease. While $bmp1a^{-/-}$ and $plod2^{-/-}$ mutants exhibited differing effects on indices of bone mass and microarchitecture, both mutants exhibited high TMD. This phenotype is consistent with the bone over-mineralization that is a hallmark of brittle bone disease (*Bishop, 2016*). While $bmp1a^{-/-}$ mutants exhibit increased TMD in all vertebral compartments (centrum, haemal arch, and neural arch), $plod2^{-/-}$ fish exhibit high TMD in the centrum only. Further, the elevation in Haem.TMD in $bmp1a^{-/-}$ mutants was focused primarily to anteriorly-located vertebrae. Phenotypic abnormalities in human OI has been shown to vary among patients and anatomical site (*Cassella et al., 1996*). The segregation of phenotypic abnormalities to different anatomical sites is likely to provide important clues into disease pathology; our workflow provides a unifying framework to systematically analyze the mechanistic basis of site-specific segregation of phenotypic abnormalities.

Our Monte Carlo simulations suggest two different cases in which analyzing vertebral phenotypic patterns via the global test confers enhanced sensitivity relative to t-tests of quantities averaged across vertebrae: (1) when the characteristic effect size of a phenotypic feature is uniformly elevated across all vertebrae, or (2) when the characteristic effect size is linearly elevated across vertebrae. In this context, of the three mutants that we analyzed in-depth ($bmp1a^{-/-}$, $plod2^{-/-}$, and *opallus*), 2 out of 3 these mutants exhibited at least one phenotypic measure that was non-uniformly across vertebrae (i.e., Haem.TMD in $bmp1a^{-/-}$ mutants, and Neur.Vol in *opallus*). For such cases, analyzing each vertebra has clear potential to provide non-redundant phenotypic information. There are other contexts in which non-uniform phenotypic abnormalities may be expected. For instance, genetic mosaicism generated within a CRISPR-based reverse genetic screen in F0 zebrafish (*Shah et al., 2015*) or an overexpression screen may manifest as different degrees of phenotypic penetrance and expressivity in each vertebral body. In this context, we are currently using both simulation as well as experimental approaches to explore the sensitivity of our workflow in discriminating mutant populations that exhibit variable phenotypic penetrance and expressivity both between and within individuals. Another instance in which non-uniform phenotypic abnormalities may arise is under environmental influences such as mechanical loading. Both non-uniform adaptation to swimming activity (*Fiaz et al., 2012*) as well as site-specific susceptibility to lordosis (*Kranenbarg et al., 2005*) have been demonstrated in the teleost spine. In unpublished studies we have expanded our analysis of zebrafish mutants whose orthologs are associated with human brittle bone disease; these studies suggest that mutations in genes that influence bone mechanical integrity may be most phenotypically penetrant in vertebrae that experience the highest mechanical loading.

Since developmental progress in zebrafish is more closely related to standard length than to age (*Parichy et al., 2009*), the interpretation of mutant phenotypes can substantially differ depending on whether mutants are compared to age-matched WT siblings (which may differ in standard length and thus developmental progress), or non-sibling WT animals matched by standard length (which could mask genetic alterations on developmental progress, and exhibit greater variation in genetics and environmental influences). In high-throughput settings where resource conservation is critical, it is not practical to dedicate resources for both experimental comparisons. Our studies demonstrate that allometric modeling is effective in transforming WT sibling data to a 'virtual' phenome scaled to the mean standard length of age-matched mutants, providing a computational means to enable both length- and age-matched fish comparisons from a single control group. Using this approach, we identified expanded centrum diameter in $plod2^{-/-}$ mutants. In mammals, cortical expansion arises due to bone remodeling, with bone formation on the periosteal surface coupled with resorption on the endosteal surface. It is unknown whether the increased centrum diameter in $plod2^{-/-}$ mutants is attributable to accelerated bone remodeling.

In addition to identifying novel phenotypes in known skeletal mutants, we also identified a novel axial skeletal mutant, *opallus*, harboring a TSHR gain-of-function mutation. Excess TSHR activity has been associated with high BMD in humans (*de Lloyd et al., 2010*). Notably, unlike $plod2^{-/-}$ and $bmp1a^{-/-}$ mutants, *opallus* exhibited high TMD in the presence of mostly normal bone mass and morphology, providing evidence of the potential for these traits to be decoupled in zebrafish mutants. More broadly, we propose that expanding our initial analyses of zebrafish mutant phenomes whose orthologs are associated with mammalian bone mass and mineral accrual is likely to facilitate the

identification of novel regulators of human bone mass, as well as identify phenotypic signatures in zebrafish that are predictive of human skeletal disease.

A future challenge is to increase the content and throughput of our approach. Since FishCuT supports the DICOM standard, it is readily ported to other microCT systems. In this context, commercial microCT systems optimized for rapid throughput imaging (e.g., through the use larger detectors to increase the number of animals per field of view, higher power x-ray sources to decrease sampling time per image, and robot-based sample changing systems) have been shown to increase imaging throughput by 10 fold or more (*Wyatt et al., 2015*). Further, Mader *et al*. described a high-throughput, fully automatic system for synchrotron-based tomographic microscopy that enabled analysis of 1300 mouse femurs (*Mader et al., 2015*). In regard to image analysis, machine learning-based approaches enable fully automated localization of boundaries of vertebral bodies in human CT/MR data (*Chu et al., 2015*), and such an approach might be used to automate seeding of segment boundaries, particularly if analysis is restricted to mutants that do not exhibit severe dysmorphic phenotypes. Finally, a long-term challenge is to extend analysis to other skeletal structures, including the craniofacial skeleton (*Pardo-Martin et al., 2013*). Notably, as microCT scans are archived, these image libraries can be retroactively analyzed as new algorithms are developed, and re-analyzed to identify new genotype-to-phenotype associations.

In conclusion, we have developed a sensitive workflow for microCT-based skeletal phenomics in adult zebrafish. Our studies provide a foundation to systematically map genotype-to-phenome relationships in zebrafish as a path to advance our understanding of the genetic basis of adult skeletal health.

## Materials and methods

### Zebrafish rearing

This study was performed in strict accordance with the recommendations in the Guide for the Care and Use of Laboratory Animals of the National Institutes of Health. All studies were performed on an approved protocol (#4306–01) in accordance with the University of Washington Institutional Animal Care and Use Committee (IACUC). Zebrafish were housed at a temperature of 28°C on a 14:10 hr light:dark photoperiod. Studies were conducted in mixed sex adult zebrafish. WT ARO and AB fish were obtained from Aquatic Research Organisms (*Recidoro et al., 2014*) and the Zebrafish International Resource Center (ZIRC, http://zebrafish.org), respectively. *opallus*$^{b1071}$ (*McMenamin et al., 2014*) and *pissarro*$^{utr8e1}$ (*Quigley et al., 2004*) were isolated in forward genetic screens. *bmp1a*$^{sa2416}$ and *plod2*$^{sa1768}$ mutant zebrafish were generated by the Zebrafish Mutation Project (ZMP) and obtained from ZIRC (*Kettleborough et al., 2013*). For all mutant lines, heterozygous mutant zebrafish were incrossed to obtain homozygous mutants. All fish were housed in plastic tanks on a commercial recirculating aquaculture system. At the desired time point, zebrafish were euthanized by MS-222 overdose or immersion in ice water. For storage, fish were either frozen at −20°C, or fixed in 4% PFA. Comparisons were only performed in fish subjected to the same storage procedure.

### MicroCT scanning

MicroCT scanning was performed using a vivaCT40 (Scanco Medical, Switzerland). Medium-resolution scans (21 µm voxel resolution) were acquired using the following settings: 55kVp, 145µA, 1024 samples, 500proj/180°, 200 ms integration time. High-resolution scans (10.5 µm voxel resolution) were acquired using the following settings: 55kVp, 145µA, 2048 samples, 1000proj/180°, 200 ms integration time. DICOM files of individual fish were generated using Scanco software, and analyzed using the custom software described below. In general, at least two fish were scanned simultaneously in each acquisition.

### Image analysis

Image processing methods were implemented as custom software developed in MATLAB (scripts were tested in v2016.a). To encourage open-source development, we implemented the MIJI package to enable calls to libraries and functions developed in FIJI/ImageJ (*Schneider et al., 2012*; *Schindelin et al., 2012*). Further, we developed a graphical user interface (GUI) to facilitate user interaction. Example DICOM images (the *bmp1a*$^{-/-}$ mutants and clutchmate controls in *Figure 5*) are

available on the Dryad data repository (http://datadryad.org/review?doi=doi:10.5061/dryad.pm41d).

Analysis consists of several stages. Stage 1 consists of preprocessing, for which we have implemented a preprocessing module in which the user is able to specify a rotation along the anteroposterior axis to orient specimens to an upright position. This module also enables slice-by-slice visualization, as well as mean or maximum intensity projections of unprocessed DICOM images.

Stage 2 consists of thresholding. In general, we have found that fish within and across clutches can exhibit significant differences in mineralization, and thus cannot be reliably analyzed using a uniform threshold value. Thus, we calculate thresholds for each animal using a semi-automatic approach. To filter out background, the user draws a ROI outlining the fish in a maximum intensity projection, all values outside this region of interest are set to 0, and the threshold is calculated using the IsoData algorithm in ImageJ (*Goeman et al., 2006*; *Goeman et al., 2004*). The threshold value may be adjusted by multiplying it by a correction factor to provide more conservative or stringent thresholding, depending on user needs. Based on a comparison of user defined thresholds and those computed using the approach described above, we multiplied the IsoData threshold by 0.73 across all experiments.

Stage 3 consists of vertebral segmentation. Our approach is to isolate individual vertebrae so that each vertebra is composed of one or more connected components that do not contain voxels from different vertebra. The user initiates planes of separation between vertebra by drawing a 'separation line' between each pair of centra. Voxels within a plane defined by the separation line are set to 0, the connected components are computed, and connected component labels are tallied for each of the two volumes separated by the plane. If the connected components with the plurality of votes in the two regions are distinct, the algorithm stops; otherwise, the separation line is extended, and the process repeated. While we found this approach robustly separated centra in all samples (including those that exhibited significant morphological deficits, see below), we encountered some cases in which ribs close to pterygiophores and associated fin rays would be considered as a single connected structure. In this case, we have implemented a manual 'cutting' tool to provide the user with the ability to sever connections between skeletal elements.

Stage 4 consists of vertebral assignment. Here, the user identifies each vertebrae's components using a user-interactive, color-coded map of connected components.

Stage 5 consists of neural arch, centrum, and haemal arch segmentation. Using a supervised algorithm, FishCuT creates 3D image masks of the three regions for each vertebral body. The 3D masks are formed in part by utilizing the user's inputs to the separation-plane growing algorithm, i.e. the endpoints of the line separating adjacent centra. We create the 3D neural arch mask as the entire region above the line extended between the two upper-most inputs for a vertebra. The centrum mask is defined by the region inside all four points for a vertebra. We include an adjustable buffer, with a default of five voxels, surrounding the centrum mask. The haemal arch mask is defined by substracting the region of the vertebrae from the regions defined by the centrum and neural arch masks. All segmentations can be visually inspected from an outputted image file containing colored-coded regions superimposed on the original image.

Stage 6 is the calculation of phenotypic features. Local thickness is computed using a model-independent method (*Hildebrand and Ruegsegger, 1997*) implemented as the Local Thickness plugin in ImageJ (*Dougherty and Kunzelmann, 2007*). Volume and surface area were computed using the *nnz* and *bwperim* functions in MATLAB. TMD was computed using the following relationship: mgHA/cm$^3$ = (x/4096)*slope + intercept, where x = the pixel intensity in the DICOM image, and the values for slope (281.706) and intercept ($-195.402$) were acquired during scanner calibration.

## Statistical analysis

For mutant fish with a known axial skeletal phenotype (*bmp1a$^{-/-}$* and *plod2$^{-/-}$*), results are reported from a single experiment; for mutant fish with a newly identified axial skeletal phenotype, results are reported from two experiments (an initial screen with n = 2 fish/group, followed by follow up analysis with n = 5 fish/group). Each biological replicate represents one technical replicate. Outliers were not identified in the study. All statistical analyses were performed in GraphPad Prism or R (*Team RC, 2015*). For comparisons of medium and high scanning resolution, linear regressions were performed with intercept set to zero. Two tailed t-tests with unequal variances were used for univariate analysis between two groups. In all other cases, multivariate analysis was performed using the globaltest

package (*Goeman et al., 2006*; *Goeman et al., 2004*). $p < 0.05$ was considered statistically significant in all cases.

## Acknowledgements

Research reported in this publication was supported by the National Institute of Arthritis and Musculoskeletal and Skin Diseases (NIAMS) of the National Institutes of Health (NIH) under Award Number AR066061. The content is solely the responsibility of the authors and does not necessarily represent the official views of the National Institutes of Health. RYK also acknowledges support from UW Royalty Research Fund Grant A88052, the University of Washington Department of Orthopedics and Sports Medicine. SKM acknowledges support from NIH K99/R00 GM105874 and NIH R03 HD091634. DMP acknowledges support from NIH R01 GM11233. PC acknowledges support from the Belgian Science Policy Office Interuniversity Attraction Poles (BELSPO-IAP) program through the project IAP P7/43-BeMGI. ZIRC is supported by NIH grant RR12546.

## Additional information

### Funding

| Funder | Grant reference number | Author |
|---|---|---|
| University of Washington | A88052 | Ronald Y Kwon |
| National Institutes of Health | AR066061 | Ronald Y Kwon |
| Belgian Science Policy Office Interuniversity Attraction Poles Program | IAP P7/43-BeMGI | Paul Coucke |
| National Institutes of Health | GM105874 | Sarah K McMenamin |
| National Institutes of Health | HD091634 | Sarah K McMenamin |
| National Institutes of Health | GM11233 | David M Parichy |

The funders had no role in study design, data collection and interpretation, or the decision to submit the work for publication.

### Author contributions

Matthew Hur, Conceptualization, Software, Formal analysis, Validation, Investigation, Visualization, Methodology, Writing—original draft, Writing—review and editing; Charlotte A Gistelinck, Formal analysis, Investigation, Writing—review and editing; Philippe Huber, Software, Writing—review and editing; Jane Lee, Marjorie H Thompson, Adrian T Monstad-Rios, Sarah K McMenamin, Andy Willaert, Investigation, Writing—review and editing; Claire J Watson, David M Parichy, Paul Coucke, Formal analysis, Writing—review and editing; Ronald Y Kwon, Conceptualization, Software, Formal analysis, Supervision, Funding acquisition, Investigation, Visualization, Methodology, Writing—original draft, Project administration, Writing—review and editing

### Author ORCIDs

Ronald Y Kwon http://orcid.org/0000-0001-9760-3761

### Ethics

Animal experimentation: This study was performed in strict accordance with the recommendations in the Guide for the Care and Use of Laboratory Animals of the National Institutes of Health. All studies were performed on an approved protocol (#4306-01) in accordance with the University of Washington Institutional Animal Care and Use Committee (IACUC).

### Decision letter and Author response

Decision letter https://doi.org/10.7554/eLife.26014.036
Author response https://doi.org/10.7554/eLife.26014.037

## Additional files

### Supplementary files

• Transparent reporting form
DOI: https://doi.org/10.7554/eLife.26014.033

### Major datasets

The following dataset was generated:

| Author(s) | Year | Dataset title | Dataset URL | Database, license, and accessibility information |
|---|---|---|---|---|
| Hur M, Gistelinck C, Huber P, Lee J, Thompson M, Monstad-Rios A, Watson C, McMenamin S, Willaert A, Parichy D, Coucke P, Kwon R | 2017 | Data from: microCT-Based Skeletal Phenomics in Zebrafish Reveals Virtues of Deep Phenotyping in a Single Organ System | http://dx.doi.org/10.5061/dryad.pm41d | Available at Dryad Digital Repository under a CC0 Public Domain Dedication |

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
