## [Decision Letter]

Thank you for submitting your article "microCT-Based Skeletal Phenomics in Zebrafish Reveals Virtues of Deep Phenotyping at the Whole-Organism Scale" for consideration by *eLife*. Your article has been favorably evaluated by Didier Stainier (Senior Editor) and three reviewers, one of whom is a member of our Board of Reviewing Editors. The reviewers have opted to remain anonymous.

The reviewers have discussed the reviews with one another and the Reviewing Editor has drafted this decision to help you prepare a revised submission.

General comments:

In general, all three reviewers felt this is an important contribution to the field, since it allows the zebrafish to be more readily used for analysis of complex phenotypes. Broadly, this falls under the category of phenomics, a rapidly growing field across multiple disciplines. The methodology and algorithms implemented here are well-executed and appear technically sound. However, the reviewers also felt that the authors tended to overstate their findings, and that they did not truly demonstrate that this could be used for high throughput analysis across many different mutants/backgrounds. Several times they refer to their approach as "whole-body" phenotyping; even with respect to the skeleton, they are not looking at the whole body, but only to a limited portion (the axial skeleton), of uniform embryological origin. They also say they are looking at "hundreds" of traits. This is only accurate if you count each aspect of each vertebra as a unique trait. For example, I would argue that tissue mineral density almost certainly is not unique for each vertebra, but rather varies systematically as one aspect of a mutant phenoptype. In addition, as detailed below, there are some questions regarding both the sensitivity and resolution of their methods, and how easily it can be generalized for new users. By addressing these concerns, we believe the paper will be greatly improved.

Essential required revisions:

1) Given the methods focus of this manuscript, the results in Figure 4 are critical to the conclusions of the study. There are several concerns regarding this figure.a) The authors utilize Monte Carlo simulations to estimate the power, sensitivity, and specificity of their method. However, important details regarding this simulation are absent. In particular, what type of probability distribution was used for the simulation? What evidence do the authors have that the measured phenotype (total TMD) follows this distribution function in wild type zebrafish? Are 3 fish per arm sufficient to support the assumption that this is the true probability distribution function, and to reliably measure it? If this cannot be justified, the authors should include additional fish in this analysis. Given that the authors describe their method as "enabling rapid (<5min/fish), whole body profiling," it is surprising that only 3 fish per arm were used for this figure. This is particularly concerning for panel 4F, in which two groups of n=2 are being sampled from a population of 3 fish.

b) While the authors devote considerable attention to total TMD, it is unclear how this test performs for other phenotypic measures. Does their method also provide increased sensitivity with similar specificity for other measures? An exploration of additional measures should be included to provide a more complete picture of the performance of this assay.

2) The authors clearly demonstrate in Figure 3 that for the second pre-caudal vertebrae there is a strong linear relationship between data acquired at high and medium resolution. In order to demonstrate how robust this relationship is across all vertebrae, the authors should provide a summary panel. For example, they could generate a plot analogous to that in Figure 2 using r-squared values instead of correlation coefficients.

3) One of the assumptions of removing allometric effects as the authors have done in Figure 7 and Figure 8 is that, as stated in Lleonart, Salat and Torres, 2000 "the available observations cover different values of X and it is recommended that sampling should cover systematically the entire range of variation of X in order to get good estimates of a and b." Related to the point discussed for Figure 4, the authors should address (and provide evidence) whether n=3 (Figure 8) is sufficient to meet this criteria. If not, additional fish should be included in order to accurately account for allometric effects.

4) The authors should provide example data along with their FishCuT source code to allow readers to better understand the method when attempting to implement it for their own data.

5) Zebrafish have the potential for use in overexpression screens, where phenotypes would exist as gradations instead of binary outcomes. Have the authors ever looked at fish with an overexpression phenotype where the phenotype is variable or incompletely penetrant? How robust is their analytic method to this type of data? This would be useful for ensuring the broad applicability of the method, but is not a requirement for publication.

---

## [Author Response]

General comments:In general, all three reviewers felt this is an important contribution to the field, since it allows the zebrafish to be more readily used for analysis of complex phenotypes. Broadly, this falls under the category of phenomics, a rapidly growing field across multiple disciplines. The methodology and algorithms implemented here are well-executed and appear technically sound. However, the reviewers also felt that the authors tended to overstate their findings, and that they did not truly demonstrate that this could be used for high throughput analysis across many different mutants/backgrounds.

In our revised manuscript we have clarified that our studies make two important contributions toward enabling rapid throughput analysis across a variety of different mutants/backgrounds: 1) developing the ability to analyze microCT images in less than 5min/fish using FishCuT, and 2) demonstrating proof-of-concept of the ability to acquire whole-spinal microCT scans at 5min/fish, if 8 fish are scanned simultaneously. However, the latter was a proof-of-concept study only, and several practical issues will need to be considered before broad application. This includes designing specimen holders with optimal animal packing (e.g., scanning 8 fish simultaneously requires the fish to be physically touching, and thus is not amenable to automatic segmentation), and characterizing potential imaging artifacts. We are actively exploring these issues, and plan to publish results from these studies in the near future.

Several times they refer to their approach as "whole-body" phenotyping; even with respect to the skeleton, they are not looking at the whole body, but only to a limited portion (the axial skeleton), of uniform embryological origin.

We have removed instances of "whole-body" and "organism-scale" phenotyping in the revised manuscript.

They also say they are looking at "hundreds" of traits. This is only accurate if you count each aspect of each vertebra as a unique trait. For example, I would argue that tissue mineral density almost certainly is not unique for each vertebra, but rather varies systematically as one aspect of a mutant phenoptype.

We now employ a more precise terminology in our revised manuscript (for instance, we now describe our approach as assessing hundreds of phenotypic measurements comprised of morphological and densitometric traits across a large number of anatomical sites). We have also included a new discussion in which we highlight experimental situations where a single trait (e.g., TMD) could non-uniformly vary across vertebrae.

Essential required revisions:1) Given the methods focus of this manuscript, the results in Figure 4 are critical to the conclusions of the study. There are several concerns regarding this figure.a) The authors utilize Monte Carlo simulations to estimate the power, sensitivity, and specificity of their method. However, important details regarding this simulation are absent. In particular, what type of probability distribution was used for the simulation? What evidence do the authors have that the measured phenotype (total TMD) follows this distribution function in wild type zebrafish?

Thank you for noting this omission. We have clarified that we used a multivariate normal distribution for our simulations. For the revised manuscript, to support the use of this distribution, we scanned a large cohort (n=16) of WT fish (all from the same clutch), and performed Royston tests for multivariate normality. We report the results from these tests in Figure 4—figure supplement 1.

Are 3 fish per arm sufficient to support the assumption that this is the true probability distribution function, and to reliably measure it? If this cannot be justified, the authors should include additional fish in this analysis.

This is an important, yet challenging, comment. We explored setting our sample size requirements to improve the accuracy of parameter estimates. Using formulas derived by Webb et al. (Am. Stat., 2010, 64(3): 257-262), for any single phenotypic measurement (assuming it is normally distributed), improving accuracy of parameter estimation by 10% with a reasonable probability (85%) would require an additional k=60 samples. As noted by Webb et al., in general, setting sample size requirements to improve accuracy of parameter estimation – with a reasonable probability – leads to alarmingly high sample sizes. Thus, we deemed that addressing this concern through increasing our sample size was impractical.

In light of this challenge, our strategy to corroborate our Monte Carlo simulations in *bmp1a*^-/-^ mutants was to perform an alternate method of testing using probability distribution functions (pdfs) that were derived completely independently of the *bmp1a*^-/-^ samples. Specifically, using the data from the n=16 WT fish analyzed in our first response to Essential required revisions 1a, we simulated a range of phenotypic abnormalities and performed tests of assay sensitivity and specificity (revised panels 4A-4G). An important virtue of our revised approach is that we are now able to simulate different phenotypic abnormalities that are not observed in *bmp1a*^-/-^ mutants, including different patterns of phenotypic alterations, and changes in phenotypic features that were unaffected by bmp1a mutation. In this context, our simulations in *bmp1a*^-/-^ mutants now serve to corroborate our main findings, rather than to function as the primary tests of assay sensitivity. Importantly, the congruence between revised panels 4A-4G and panels 4H-4I, whose pdfs were derived from independent samples, suggests that findings from each approach are valid.

Given that the authors describe their method as "enabling rapid (<5min/fish), whole body profiling," it is surprising that only 3 fish per arm were used for this figure. This is particularly concerning for panel 4F, in which two groups of n=2 are being sampled from a population of 3 fish.

In the original manuscript, the primary purpose of panel 4F was to provide a secondary method to corroborate Monte Carlo simulations in *bmp1a*^-/-^ mutants. In this context, our new studies performed in our second response to Essential required revisions 1a now make panel 4F somewhat redundant. Since panel 4F employed methods that were less rigorous and somewhat unconventional, we have decided to remove these data from the revised manuscript. However, we are prepared to re-add these data as a supplementary figure if requested.

b) While the authors devote considerable attention to total TMD, it is unclear how this test performs for other phenotypic measures. Does their method also provide increased sensitivity with similar specificity for other measures? An exploration of additional measures should be included to provide a more complete picture of the performance of this assay.

As described in our first response to Essential required revisions 1a, we have revised our approach to enable evaluation of sensitivity across all phenotypic features and for multiple phenotypic patterns. In revised Figure 4 we describe assay performance across multiple features; a comprehensive summary of assay performance is now included as part of Figure 4—figure supplement 2.

2) The authors clearly demonstrate in Figure 3 that for the second pre-caudal vertebrae there is a strong linear relationship between data acquired at high and medium resolution. In order to demonstrate how robust this relationship is across all vertebrae, the authors should provide a summary panel. For example, they could generate a plot analogous to that in Figure 2 using r-squared values instead of correlation coefficients.

In the revised manuscript we have included results for multiple vertebrae (Figure 3—figure supplement 1), and created heat maps as suggested (Figure 3—figure supplement 2). Note that we were unable to extend our analysis to all vertebrae due to the fact that acquiring whole-body scans at high-resolution is computationally prohibitive. However, in the three vertebrae we analyzed, we found virtually identical results, suggesting our results are generalizable.

3) One of the assumptions of removing allometric effects as the authors have done in Figure 7 and Figure 8 is that, as stated in Lleonart, Salat and Torres, 2000 "the available observations cover different values of X and it is recommended that sampling should cover systematically the entire range of variation of X in order to get good estimates of a and b." Related to the point discussed for Figure 4, the authors should address (and provide evidence) whether n=3 (Figure 8) is sufficient to meet this criteria. If not, additional fish should be included in order to accurately account for allometric effects.

In the revised manuscript we have clarified that once estimates of a and b are obtained in a comprehensive sample of WT animals, they can be applied to other WT fish in experimental groups of an arbitrary sample size, such as the n=3 WT fish in Figure 8. In the original manuscript, a and b were estimated from n=12 fish. In the revised manuscript, we have analyzed additional fish to increase our sample size to n=16 (Figure 7—figure supplement 1), providing a more uniform sampling over different standard lengths. Finally, we have included new convergence analyses suggesting that our use of n=16 samples was sufficient to provide reliable model parameter estimates.

4) The authors should provide example data along with their FishCuT source code to allow readers to better understand the method when attempting to implement it for their own data.

We have included example microCT data by uploading DICOM files to Dryad (data will be published at time of paper acceptance). We have also uploaded an updated version of FishCuT with increased functionality that includes the ability to analyze TIFF images to GitHub.

5) Zebrafish have the potential for use in overexpression screens, where phenotypes would exist as gradations instead of binary outcomes. Have the authors ever looked at fish with an overexpression phenotype where the phenotype is variable or incompletely penetrant? How robust is their analytic method to this type of data? This would be useful for ensuring the broad applicability of the method, but is not a requirement for publication.

We have included a new discussion addressing these questions.